# In vivo partial reprogramming of myofibers promotes muscle regeneration by remodeling the stem cell niche

Chao Wang[1], Ruben Rabadan Ros[1], Paloma Martinez-Redondo[1], Zaijun Ma[1], Lei Shi[1], Yuan Xue[1], Isabel Guillen-Guillen[1], Ling Huang[2], Tomoaki Hishida[1], Hsin-Kai Liao[1], Estrella Nuñez Delicado [3], Concepcion Rodriguez Esteban[1], Pedro Guillen-Garcia[4], Pradeep Reddy [1] & Juan Carlos Izpisua Belmonte [1✉]

Short-term, systemic expression of the Yamanaka reprogramming factors (*Oct-3/4*, *Sox2*, *Klf4* and *c-Myc* [OSKM]) has been shown to rejuvenate aging cells and promote tissue regeneration in vivo. However, the mechanisms by which OSKM promotes tissue regeneration are unknown. In this work, we focus on a specific tissue and demonstrate that local expression of OSKM, specifically in myofibers, induces the activation of muscle stem cells or satellite cells (SCs), which accelerates muscle regeneration in young mice. In contrast, expressing OSKM directly in SCs does not improve muscle regeneration. Mechanistically, expressing OSKM in myofibers regulates the expression of genes important for the SC microenvironment, including upregulation of p21, which in turn downregulates *Wnt4*. This is critical because Wnt4 is secreted by myofibers to maintain SC quiescence. Thus, short-term induction of the Yamanaka factors in myofibers may promote tissue regeneration by modifying the stem cell niche.

[1] Gene Expression Laboratory, Salk Institute for Biological Studies, La Jolla, CA, USA. [2] Integrative Genomics and Bioinformatics Core, Salk Institute for Biological Studies, La Jolla, CA, USA. [3] Universidad Católica San Antonio de Murcia (UCAM), Campus de los Jerónimos, Guadalupe, Spain. [4] Department of Traumatology and Research Unit, Clinica CEMTRO, Madrid, Spain. ✉email: belmonte@salk.edu

Reprogramming of somatic cells to a pluripotent state by overexpressing the Yamanaka factors (Oct-3/4, Sox2, Klf4, and c-Myc [OSKM]) is a long and complex process[1–3]. Cellular reprogramming is widely utilized for disease modeling in vitro[4]. However, reprogramming in vivo induces tumor development[5–7]. Our lab showed that partial reprogramming by short-term expression of reprogramming factors ameliorated aging hallmarks without tumor formation[8], opening a possible application of this approach in vivo. Recently, other reports have demonstrated rejuvenation of dentate gyrus cells, retinal ganglion cells, chondrocytes, and muscle stem cells using reprogramming factors[9–11], reinforcing its potential application in clinical settings. Besides amelioration of cellular aging hallmarks, reprogramming factors promote tissue regeneration in aged mice[8,10]. However, it is unknown whether OSKM-improved regeneration is solely a result of its rejuvenating effect.

Muscle regeneration is primarily mediated by muscle stem cells or satellite cells (SCs), which reside in a characteristic niche located between the basal lamina and plasma membrane of myofibers[12]. The regenerative capacity of SCs is influenced by both intrinsic modulators and the extrinsic microenvironment[13–15]. We have shown that partial reprogramming promotes skeletal muscle regeneration in 12-month-old mice[8], but these studies were performed by expressing OSKM systemically (i.e., in all cell types)[8]. It is therefore unclear whether intrinsic or niche-specific factors contributed to the observed improvement in muscle regeneration.

In this work, we generate myofiber- and SC-specific OSKM induction mouse models to investigate the effect of OSKM induction on extrinsic and intrinsic modulators of SCs, respectively. In addition, we chose young mice to investigate whether the improvement of regeneration can be achieved by OSKM induction regardless of its rejuvenating effect. Our data shows that myofiber-specific OSKM induction accelerates muscle regeneration through downregulating the myofiber-secreted niche factor, Wnt4, to induce the activation and proliferation of SCs. In contrast, SC-specific OSKM induction does not improve muscle regeneration in young mice. We conclude that partial reprogramming via OSKM can remodel the SC niche to induce SC activation and proliferation and accelerate muscle regeneration.

## Results

**RNA-seq analysis of myofiber-specific OSKM induction mice.**
To investigate the effect of OSKM on the SC niche, we crossed Acta1-Cre mice to mice carrying an OSKM polycistronic cassette (4F) plus a LoxP-STOP-LoxP cassette-blocked rtTA *trans*-activator and GFP transgenes. In this mouse model (Acta1-Cre/4F$^{het}$), the LoxP-STOP-LoxP cassette is excised by Cre in myofibers, resulting in rtTA and GFP expression. Doxycycline (Dox) administration induces OSKM expression specifically in myofibers (Fig. 1a). Immunostaining verified the expression of Sox2 and GFP in extensor digitorum longus (EDL) muscles of Acta1-Cre$^+$/4F$^{het}$ mice but not in Acta1-Cre$^-$/4F$^{het}$ mice after introduction of Dox (1 mg/ml) in their drinking water for 2.5 days (Supplementary Fig. 1). Then, 2-month-old Acta1-Cre$^+$/4F$^{het}$ and Acta1-Cre$^-$/4F$^{het}$ mice (controls) were administered Dox for 2.5 or 8.5 days to induce OSKM. We isolated the EDL and soleus (SOL) muscles, which are predominantly fast- and slow-twitch myofibers, respectively. RNA was then extracted and subjected to RNA-sequencing (RNA-seq) analysis to determine global changes in gene expression (Fig. 1b). We first sought to confirm OSKM expression, and indeed measured higher levels of OSKM in Cre$^+$ EDL muscles compared with Cre$^-$ EDL muscles (Fig. 1c). In SOL muscles, however, Oct4, Sox2, and c-Myc were induced, but the expression of Klf4 was similar between Cre$^+$ and Cre$^-$ SOL

muscles (Fig. 1d). In addition, the expression of Oct4, Sox2, and c-Myc was 4-fold higher in Cre$^+$ EDL muscles compared with Cre$^+$ SOL muscles (baseline expression of these genes was comparable between Cre$^-$ EDL and Cre$^-$ SOL muscles) (Fig. 1c, d). These results indicate that OSKM was more strongly induced in fast-twitch muscles than in slow-twitch muscles.

Principal component analysis (PCA) of the RNA-seq data revealed a clear separation between Cre$^+$ and Cre$^-$ samples, and the separation distance was wider in EDL muscles than in SOL muscles (Fig. 1e), consistent with the levels of OSKM induction. We then pooled all differentially expressed (DE) genes from both EDL and SOL muscles and used them as input for pattern analysis. Eleven modules were identified, but only three were enriched for GO terms associated with biological processes (Supplementary Fig. 2a, b). Each of these three modules contained genes repressed or activated by OSKM induction in both EDL and SOL muscles (Supplementary Fig. 2a). As with the PCA analysis, the gene expression level induced or repressed by OSKM was more pronounced in EDL muscles than in SOL muscles (Supplementary Fig. 2a). The repressed genes were enriched for pathways related to metabolism and muscle differentiation (Supplementary Fig. 2b). Activated genes were enriched in pathways related to cytoskeleton organization (Supplementary Fig. 2b). These results suggest that fast-twitch muscles are a more appropriate model for evaluating the effects of OSKM on muscle regenerative capacity.

We thus further analyzed the RNA-seq data obtained from EDL muscles to determine the time course of gene expression changes following OSKM induction. First, levels of OSKM upregulation were comparable between 2.5 and 8.5 days of Dox administration (Fig. 1c). In addition, 2.5 and 8.5 days of Dox treatment induced a similar Euclidean distance between the Cre$^+$ and Cre$^-$ samples, consistent with the PCA analysis (Fig. 1e, f). Gene set enrichment analysis (GSEA) showed that 2.5 and 8.5 days of treatment affected the same genetic pathways (Supplementary Fig. 3a). Furthermore, DE genes resulting from 2.5 or 8.5 days of treatment had similar distribution patterns when visualized using volcano plots (Fig. 1g). Interestingly, more genes were differentially expressed following 2.5 days of OSKM induction than seen following 8.5 days of treatment (Fig. 1g). A total of 1142 genes were upregulated and 697 genes were downregulated only in the 2.5-day treatment group (Fig. 1h). In contrast, 181 and 202 genes were upregulated and downregulated, respectively, only in the 8.5-day treatment group (Fig. 1h). These 202 downregulated genes were enriched in GO biological process terms related to cellular response to stimulus (Supplementary Fig. 3b), whereas the 181 upregulated genes were not enriched for a specific GO biological process term. The 2.5-day and 8.5-day treatment regimens shared 876 upregulated and 578 downregulated genes (Fig. 1h). The GO biological process term most enriched in these upregulated genes was related to cytoskeleton organization, whereas downregulated genes were related to oxidative phosphorylation (Fig. 1h). Based on these results, we used a short-term OSKM induction protocol for all subsequent studies.

**OSKM induction in myofibers accelerates muscle regeneration.**
Although changes in gene expression were observed in myofibers following short-term OSKM induction, it was still unclear whether partial reprogramming of myofibers would affect muscle regeneration. We therefore studied the effect of myofiber-specific OSKM induction on muscle regeneration using a previously established cyclical and short-term OSKM induction protocol. Dox was administered for 2 days followed by 5 days of Dox withdrawal, and then the cycle was repeated[8]. We used the tibialis anterior (TA) muscle, another fast-twitch muscle, for these

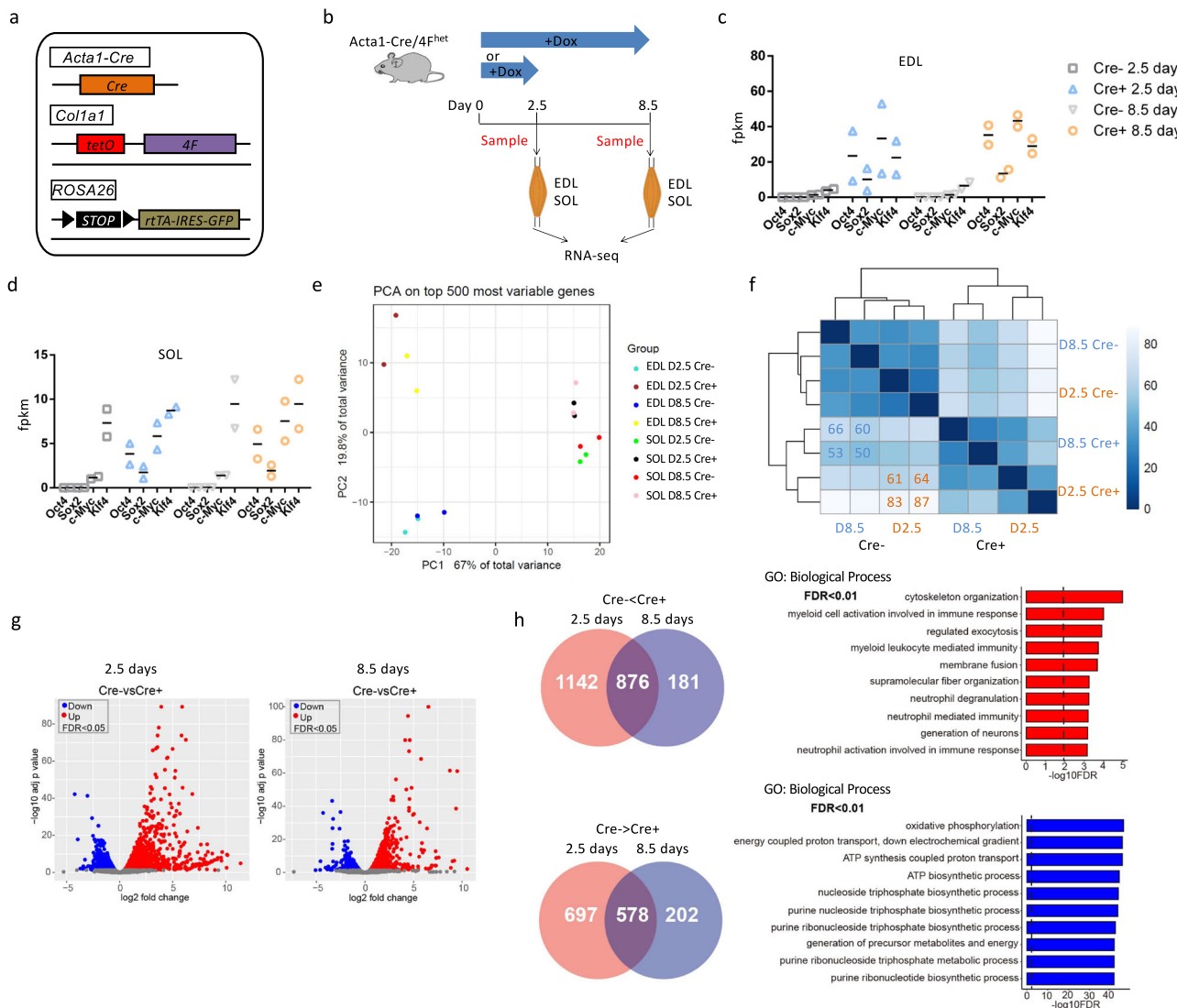

**Fig. 1 RNA-seq analysis of slow-twitch and fast-twitch muscles with myofiber-specific OSKM induction. a** Illustration of the genetic makeup for myofiber-specific OSKM inducible mouse model. For simplicity, Cre⁺ represents Acta1-Cre⁺/4Fʰᵉᵗ and Cre⁻ represents Acta1-Cre⁻/4Fʰᵉᵗ mice. **b** Schematic representation of the experimental design. **c** The fpkm of OSKM in Cre⁻ and Cre⁺ EDL muscles. $n = 2$ biological independent samples. **d** The fpkm of OSKM in Cre⁻ and Cre⁺ SOL muscles. $n = 2$ biological independent samples. **e** Principal component analysis on top 500 most variables genes of Cre⁻ and Cre⁺ EDL and SOL muscles. **f** Euclidean distance analysis of the transcriptome of Cre⁻ and Cre⁺ EDL muscles. **g** Volcano plots of DE genes in EDL muscles after 2.5- or 8.5-days Dox treatment. **h** Enriched GO terms in the biological process of commonly changed genes after 2.5- or 8.5-days Dox treatment.

analyses. OSKM expression was induced in the TA muscle following 2 days of Dox treatment, with OSKM levels returning back to control levels following 5 days of withdrawal (Fig. 2a). Immunostaining confirmed the specific expression of Oct4 in nuclei of myofibers but not in Pax7⁺ (the SCs marker) cells after 2 days of Dox treatment (Fig. 2b). After 3 cycles of OSKM induction, Acta1-Cre/4Fʰᵉᵗ mice (2–4 months old) were treated with cardiotoxin (CTX), which led to TA muscle degeneration. Immediately after CTX injury, the mice were administrated drinking water containing bromodeoxyuridine (BrdU) for 1 day to label cycling SCs (Fig. 2c). Muscle regenerative capacity was evaluated 3 days post CTX injection (Fig. 2c). At the early stage of muscle regeneration, newly regenerated immature myofibers are marked by embryonic myosin heavy chain (eMHC). Cre⁺ muscles had more eMHC⁺ myofibers compared to Cre⁻ muscles (Fig. 2d), indicating an earlier regenerative process in Cre⁺ muscles. In addition, Cre⁺ samples had more Pax7⁺ cells than Cre⁻ samples (Fig. 2e). Co-staining of Pax7 and BrdU did not

show a difference in the percentage of Pax7⁺BrdU⁺ cells, as all SCs entered into the cell cycle at the early stage of regeneration (Fig. 2e). We further evaluated muscle regeneration capacity 6 days post CTX injury (Fig. 2f). Regenerating myofibers were identified by central nuclei and immature myofibers were marked by eMHC. Cre⁺ muscles had a lower percentage of eMHC⁺ regenerating myofibers compared to Cre⁻ muscles (Fig. 2g), indicating a higher percentage of mature regenerating myofibers in Cre⁺ muscles. Compared to mature eMHC⁻ myofibers, eMHC⁺ immature myofibers had a smaller myofiber size (Fig. 2g). Consistently, regenerated myofibers in Cre⁺ muscles were larger than those observed in Cre⁻ muscles, with more myofibers that were 30–40 μm and fewer that were 10–20 μm in diameter (Fig. 2h). The expression of MyoD distinguishes Pax7⁺ MyoD⁺ proliferating SCs (often referred as myoblasts) from Pax7⁺MyoD⁻ quiescent SCs. Cre⁺ samples had more Pax7⁺ MyoD⁺ myoblasts than Cre⁻ samples (Fig. 2i). In contrast, the number of Pax7⁺MyoD⁻ SCs was comparable in Cre⁻ and Cre⁺

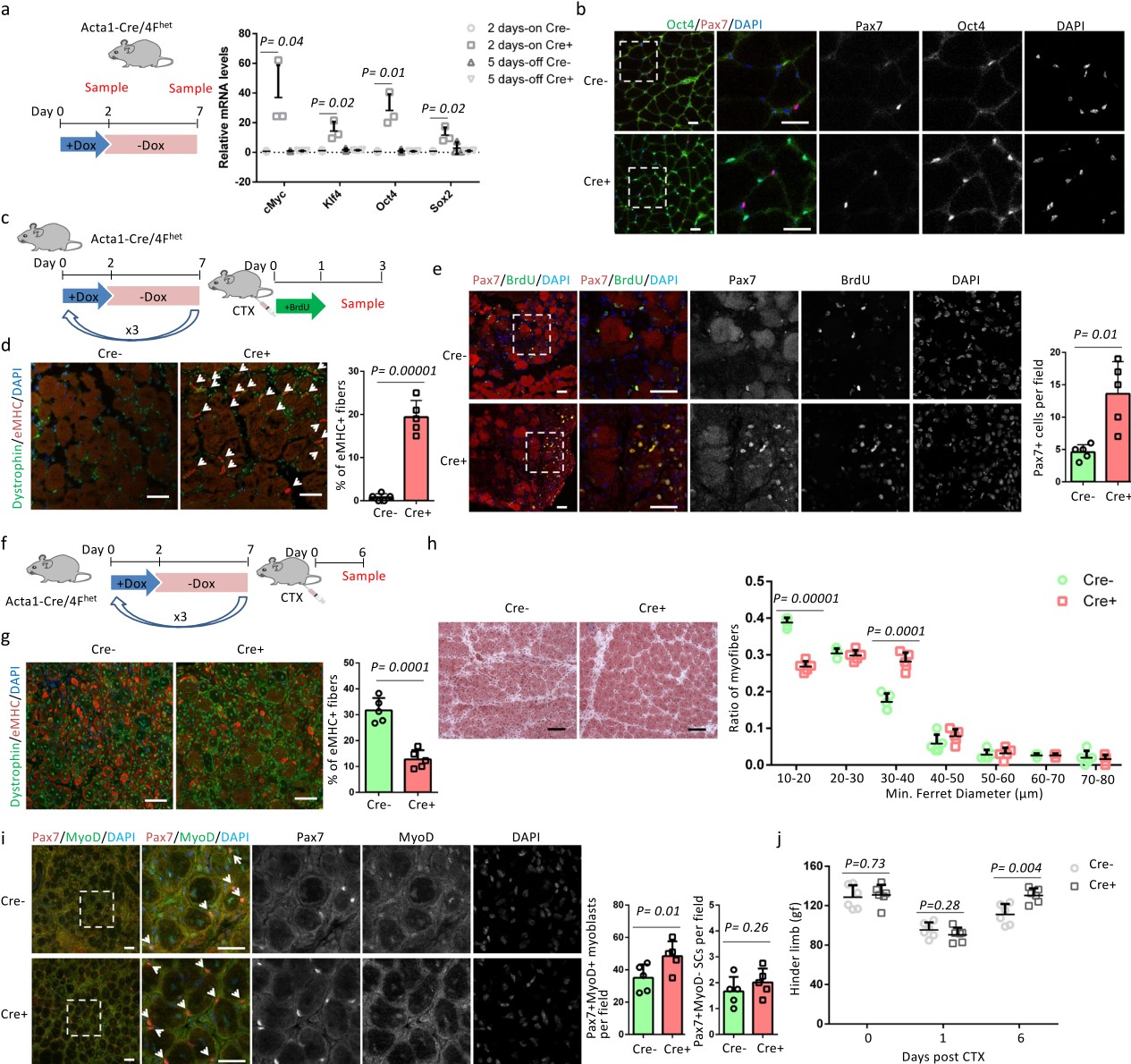

**Fig. 2 Myofiber-specific OSKM induction accelerates muscle regeneration. a** Relative mRNA levels of OSKM in TA muscles after 2-days Dox treatment (2 days-on) and after 5-days Dox withdrawal (5 days-off). n = 3 Cre− mice for 2 days-on and 5 days-off. n = 3 Cre+ mice for 2 days-on and 4 Cre+ mice for 5 days-off. Error bars represent mean+SD. **b** Immunostaining of Oct4 and Pax7 in TA muscle sections. Scale bars = 50 μm. Similar results were repeated independently in four pairs of mice. In addition, similar results were repeated three times for each mouse. **c, f** Schematic representation of the experimental design. **d, g** Immunostaining of embryonic myosin heavy chain (eMHC) and Dystrophin in TA muscle sections, and the quantification of the percentage of immature myofibers that express eMHC. Error bars represent mean + SD of five biological independent samples. Arrows indicate eMHC+ myofibers in panel **d**. Scale bars = 50 μm. **e** Immunostaining of Pax7 and BrdU, and quantification of Pax7 in TA muscle sections. Error bars represent mean + SD of five mice. Representative regions are shown at higher magnification. Scale bars = 50 μm. **h** H&E staining of TA muscle sections and myofiber size distributions in TA muscle sections. Error bars represent mean + SD of five mice. Scale bars = 100 μm. **i** Immunostaining and quantification of Pax7 and MyoD in TA muscle sections. Representative regions are shown at higher magnification. Pax7+ cells are indicated by arrows. Scale bars = 50 μm. Error bars represent mean + SD of five mice. **j** Hinder limb grip strength at different time points after CTX injection. Error bars represent mean + SD of six mice. A two-sided unpaired Student's t-test was performed.

samples (Fig. 2i), indicating Cre+ samples had an expansion of myoblasts without changing the SCs number. Next, we performed three cycles of OSKM induction followed by injury to the gastrocnemius (GA) and TA muscles of Acta1-Cre/4Fhet mice with CTX. The grip strength of hind limbs was measured 1 and 6 days post CTX treatment. Whereas Cre+ and Cre− hind limbs had similarly low grip strength 1 day post CTX injection, Cre+ muscles had higher grip strength than Cre− muscles 6 days post CTX treatment (Fig. 2j), indicating that Cre+ muscles

restored their function faster than Cre− muscles. These results suggest that partial reprogramming of myofibers accelerated muscle regeneration.

To investigate whether muscle regeneration could be accelerated with OSKM after CTX treatment, rather than treatment prior to injury, we used the TA muscles of Acta1-Cre/4Fhet mice. Immediately after CTX treatment, the mice were administrated Dox for 2.5 days (Supplementary Fig. 4a). Muscle regeneration capacity was evaluated 6 days post CTX injury (Supplementary

Fig. 4a). The percentages of eMHC+ regenerating myofibers were comparable between Cre+ and Cre− muscles (Supplementary Fig. 4b, c). The myofiber sizes were also similar between Cre+ and Cre− muscles (Supplementary Fig. 4d). In addition, there was no difference in the number of Pax7+ cells (Supplementary Fig. 4e). These data show that OSKM induction after muscle injury could not accelerate muscle regeneration, reinforcing the benefits of preinjury induction of OSKM.

**OSKM induction in myofibers promotes SC proliferation.** To understand how partial reprogramming of myofibers may accelerate SC-mediated muscle regeneration, we analyzed the state and number of SCs just before muscle injury. Acta1-Cre/4F^het mice were treated with the same 2 days-on/5 days-off cyclic induction protocol. After the second and third rounds of Dox administration, BrdU was added to the drinking water for 2 days to label cycling SCs (Fig. 3a). Co-staining these muscles for BrdU and Pax7 showed that Cre+ muscles had a 3-fold higher percentage of Pax7+BrdU+ cells than Cre− muscles, indicating that myofiber-specific expression of OSKM promoted the proliferation of SCs (Fig. 3b, c). As a consequence, Cre+ muscles had more Pax7+ cells than Cre− muscles (Fig. 3d, e). The expression of MyoD marks the activation of SCs. Co-staining for Pax7 and MyoD showed that Cre+ muscles had a higher percentage of activated (MyoD+) SCs and a lower percentage of quiescent (MyoD−) SCs than Cre− muscles (Fig. 3d, f). The percentage of SCs-derived myocytes (MyoD+Pax7−) was comparable between Cre+ and Cre− muscles (Fig. 3d, f). H&E staining did not reveal any morphological differences between Cre+ and Cre− muscle sections and the myofibers in Cre− and Cre+ muscles lacked central nuclei (Supplementary Fig. 5a). In addition, immunostaining showed that there are no eMHC+ myofibers in Cre− and Cre+ muscles before an injury (Supplementary Fig. 5b), indicating that the activated SCs did not spontaneously contribute to myofibers of Cre+ muscles. To exclude the possibility that the increased number of Pax7+ cells were derived from myofibers through dedifferentiation, Acta1-Cre+/4F^het mice were bred with Ai14 (Rosa-LoxP-STOP-LoxP-tdTomato)^homo mice to generate Acta1-Cre+/4F^het/Ai14^het mice, where myofibers and their lineage were labeled with Tdtomato (Td). All the Pax7+ cells were Td−, indicating that Pax7+ cells were not derived from myofibers (Supplementary Fig. 5c). Together, these results suggest that partial reprogramming of myofibers induced the activation and proliferation of SCs.

We next isolated single myofibers from Acta1-Cre/4F^het mice treated with Dox for 2.5 days (Fig. 3g). Cre+ myofibers were associated with 2-fold more Pax7+ cells than Cre− myofibers (Fig. 3h, i). Moreover, the percentage of activated (MyoD+) SCs was 3-fold higher in Cre+ myofibers than Cre− myofibers (Fig. 3h, j). Conversely, the percentage of quiescent (MyoD−) SCs was decreased by half in Cre+ myofibers compared to Cre− myofibers, and there was no difference in the percentage of MyoD+Pax7− cells (Fig. 3h, j). As expected, we found SC doublets, indicative of SC proliferation, only in Cre+ myofibers (Supplementary Fig. 5d). We next cultured these myofibers for 3 days (Fig. 3k). The number of cell clusters was significantly higher in Cre+ myofibers than Cre− myofibers (Fig. 3k, l). Additionally, all Pax7+ cells expressed MyoD in both Cre− and Cre+ myofibers (Fig. 3k). The number of Pax7+MyoD+ myoblast per cluster was higher in Cre+ myofibers (Fig. 3k, m), excluding the possibility that the activated SCs exited the cell cycle for premature differentiation. Collectively, these results prove that partial reprogramming of myofibers induced the activation and proliferation of SCs and the expansion of myoblasts.

**OSKM induction in myofibers does not change SC self-renewal.** Self-renewal of SCs is essential for the long-term maintenance of skeletal muscles. To investigate whether self-renewal capacity was affected by OSKM induction, we first analyzed the state and number of SCs 14 and 30 days after TA muscle injury (Fig. 4a, g). Co-staining of MyoD and Pax7 showed that Cre+ muscles had more Pax7+ cells compared to Cre− muscles 14 days post CTX injection (Fig. 4b, c). The percentage of Pax7+MyoD− SCs, which are capable of self-renewal was comparable between Cre+ and Cre− muscles (Fig. 4b, c). We further utilized eMHC staining, Mallory's trichrome staining and H&E staining to evaluate muscle regeneration. Cre+ muscles had a lower percentage of eMHC+ regenerating myofibers compared to Cre− muscles (Fig. 4d). Mallory's trichrome staining showed that the Cre+ muscles had less fibrosis compared to Cre− muscles (Fig. 4e). In addition, regenerated myofibers in Cre+ muscles were larger than those observed in Cre− muscle, with more myofibers that were 50–60 μm in diameter (Fig. 4f). These results further confirm that myofiber-specific OSKM induction accelerated muscle regeneration. At 30 days post-injury, the number of Pax7+ cells and the percentage of Pax7+MyoD− SCs were comparable between Cre+ and Cre− muscles (Fig. 4h, i). Muscle structure was restored and there were no eMHC+ myofibers in Cre+ and Cre− muscles (Fig. 4j). Fibrosis was similarly low in Cre+ and Cre− muscles (Fig. 4k), and myofiber sizes were comparable between Cre+ and Cre− muscles (Fig. 4l). These results show that the SC pool was maintained after OSKM induction.

To further evaluate the self-renewal capability of SCs after cyclic OSKM induction, we performed CTX injury three times with an interval of 7 days between each injection (Fig. 5a). In case the self-renewal of SCs was inhibited, the serial injury would exaggerate the depletion of SCs and reduce muscle regeneration capability. At 7 days after the last CTX injury (day 21), there was no difference in the number of Pax7+ cells between Cre+ and Cre− muscles (Fig. 5b). We did eMHC immunostaining and H&E staining to evaluate muscle regeneration capability. The percentage of eMHC+ regenerating myofibers and myofiber sizes were comparable between Cre+ and Cre− muscles (Fig. 5c, d). We also checked the number and state of Pax7+ cells 30 days post the last CTX injury. BrdU administration was included to label activated SCs or those that proliferated after the last CTX injection (Fig. 5e). The number of Pax7+ cells was comparable between Cre+ and Cre− muscles (Fig. 5f). Pax7+MyoD−BrdU+ SCs were derived from activated or proliferated SCs. The percentage of Pax7+MyoD−BrdU+ SCs was also similar in Cre+ and Cre− muscles. Together, these data prove that myofiber-specific OSKM induction did not change the self-renewal capacity of SCs.

**OSKM induction in SCs does not change muscle regeneration.** We next expressed OSKM directly in SCs and measured the effect on SCs and muscle regeneration. For this purpose, we generated SC-specific 4F mice (Pax7^creER/4F^het), which carries Pax7^creER, tetO-4F, and LoxP-STOP-LoxP-rtTA. Pax7^creER/4F^het mice were administered Tamoxifen (TMX) for 5 days and then treated with the same cyclical OSKM induction protocol that Acta1-Cre/4F^het mice received. Real-time PCR using whole muscle lysates confirmed that OSKM was induced immediately after Dox treatment (Supplementary Fig. 6a). Young (6–8 weeks old) Pax7^creER/4F^het mice were treated with Dox and then administered BrdU following the second and third rounds of Dox treatment (Supplementary Fig. 6b). We labeled muscle sections for Pax7, MyoD, and BrdU, and analyzed the number and state of SCs (Supplementary Fig. 6c, d). The number of Pax7+ cells, as well as the percentages of MyoD+ and BrdU+ SCs were all comparable between Cre+ and Cre− muscles (Supplementary Fig. 6e–g), indicating that SC-specific OSKM induction did not lead to the

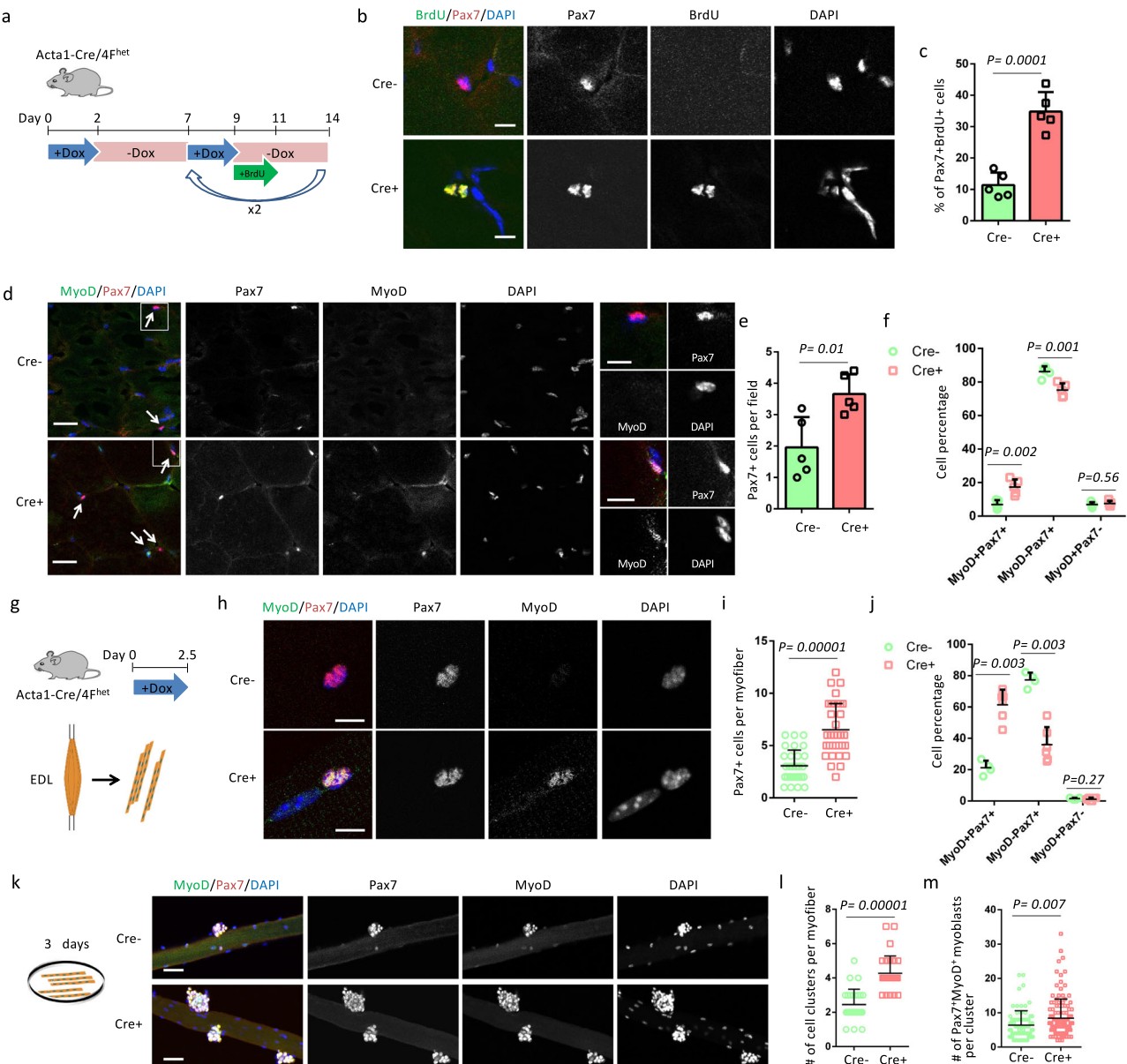

**Fig. 3 Myofiber-specific OSKM induction promotes the activation and proliferation of satellite cells. a** Schematic representation of the experimental design. **b** Immunostaining of Pax7 and BrdU in TA muscle sections. Scale bars = 10 μm. **c** Quantification of the percentage of SCs with BrdU signals. Error bars represent mean + SD of five mice. **d** Immunostaining of Pax7 and MyoD in TA muscle sections. Pax7$^+$ cells are indicated by arrows. Scale bars = 50 μm. Representative regions are shown at higher magnification with Scale bars = 10 μm. **e** Quantification of Pax7$^+$ cells per field. Error bars represent mean + SD of five mice. **f** Quantification of the percentage of Pax7$^+$MyoD$^-$, Pax7$^+$MyoD$^+$, and Pax7$^-$MyoD$^+$ cells. Error bars represent mean + SD of five mice. **g** Illustration of the design for myofibers analysis. **h** Immunostaining of Pax7 and MyoD in single myofibers. Scale bars = 10 μm. **i** Quantification of Pax7$^+$ cells per myofiber. n = 31 Cre− myofibers and 35 Cre+ myofibers, respectively. Error bars represent mean + SD. **j** Quantification of the percentage of Pax7$^+$MyoD$^-$, Pax7$^+$MyoD$^+$, and Pax7$^-$MyoD$^+$ cells in single myofibers. Error bars represent mean + SD of four Cre− EDL muscles and 6 Cre+ EDL muscles. **k** Immunostaining of Pax7 and MyoD in single myofibers after culture for 3 days. Scale bars = 50 μm. **l** Quantification of cell clusters per myofiber. n = 28 Cre− myofibers and 30 Cre+ myofibers, respectively. Error bars represent mean + SD. **m** Quantification of Pax7$^+$MyoD$^+$ myoblasts per cluster. n = 76 cell clusters on Cre− myofibers and 128 cell clusters on Cre+ myofibers. Error bars represent mean + SD. A two-sided unpaired Student's t-test was performed.

activation and proliferation of SCs. We then examined the effect of SC-specific OSKM induction on muscle regeneration. Two-month-old Pax7$^{creER}$/4F$^{het}$ mice were treated with cyclical Dox followed by the injection of CTX into TA muscles (Supplementary Fig. 6 h). Six days post-injury, no differences in the morphology or size of regenerated myofibers were observed between Cre$^+$ and Cre$^-$ muscles (Supplementary Fig. 6 h). In addition, the number of myoblasts in regenerated muscles was comparable between Cre$^+$ and Cre$^-$ muscles (Supplementary Fig. 6i).

To understand why OSKM did not result in SC proliferation in Pax7$^{creER}$/4F$^{het}$ mice, SC-derived myoblasts were isolated from Pax7$^{creER}$/4F$^{het}$ mice and treated with Dox (1 μg/ml) for 2 days to examine the effect of OSKM induction on MyoD, which enhances myoblasts proliferation by activating miR-133[16,17]. We confirmed that OSKM was induced in Cre$^+$ myoblasts compared to Cre$^-$ myoblasts (Supplementary Fig. 7a). OSKM inhibited the expression of MyoD and miR-133a (Supplementary Fig. 7b), which is consistent with the finding that Oct4 represses the expression of

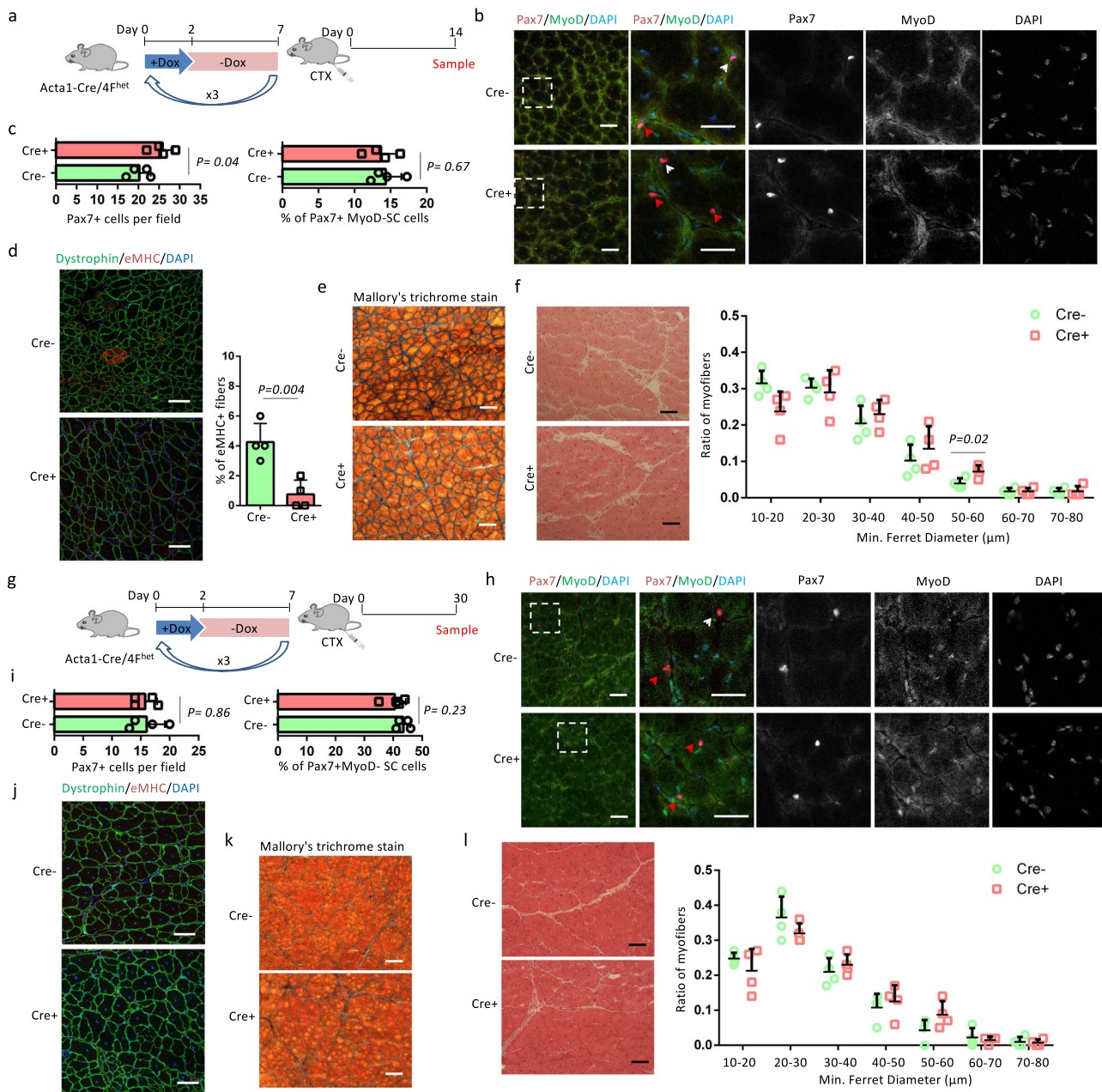

**Fig. 4 Analysis of the state and number of SCs 14 and 30 days after CTX injection. a, g** Schematic representation of the experimental design. **b, h** Immunostaining of Pax7 and MyoD in TA muscle sections. Scale bars = 50 μm. Representative regions are shown at higher magnification. Scale bars = 25 μm. White arrows indicate Pax7⁺MyoD⁺ myoblasts and Red arrows indicate Pax7⁺MyoD⁻ SCs. **c, i** Quantification of Pax7 and the percentage of Pax7⁺MyoD⁻ SCs in TA muscle sections. **d, j** Immunostaining of embryonic myosin heavy chain (eMHC) and Dystrophin in TA muscle sections, and the quantification of the percentage of immature myofibers that express eMHC. Scale bars = 50 μm. Similar results were repeated independently in four pairs of mice. In addition, similar results were repeated three times for each mouse. **e, k** Mallory's trichrome stain of TA muscle sections. Scale bars = 100 μm. Similar results were repeated independently in four pairs of mice. In addition, similar results were repeated three times for each mouse. **f, l** H&E staining of TA muscle sections and myofiber size distributions in TA muscle sections. Scale bars = 100 μm. **a–f** represents the analysis of SCs 14 days after CTX injection, and **g–l** represent the analysis of SCs 30 days after CTX injection. Error bars represent mean + SD of four mice. A two-sided unpaired Student's *t*-test was performed.

MyoD[18]. Based on these observations, we conclude that the cyclic induction of OSKM in young SCs did not improve muscle regeneration.

**OSKM induction promotes proliferation of SCs in aging mice.** To determine whether the effects of myofiber-specific OSKM induction on SCs are preserved in aging mice, we investigated the status of SCs in 15-month-old myofiber-specific 4F mice. Acta-

Cre/4Fhet mice were administrated with 3 cycles of Dox. After the second and third rounds of Dox administration, BrdU was added to the drinking water for 2 days to label cycling SCs (Supplementary Fig. 8a). Cre⁺ muscles had 1.8-fold more Pax7⁺ cells and 2.8-fold more activated (MyoD⁺) SCs than Cre⁻ muscles (Supplementary Fig. 8b-d). Co-staining of BrdU and Pax7- showed that Cre⁺ muscles had a 3-fold higher percentage of BrdU⁺ SCs than Cre⁻ muscles (Supplementary Fig. 8e, f). We then studied muscle regeneration capacity 7 days post CTX

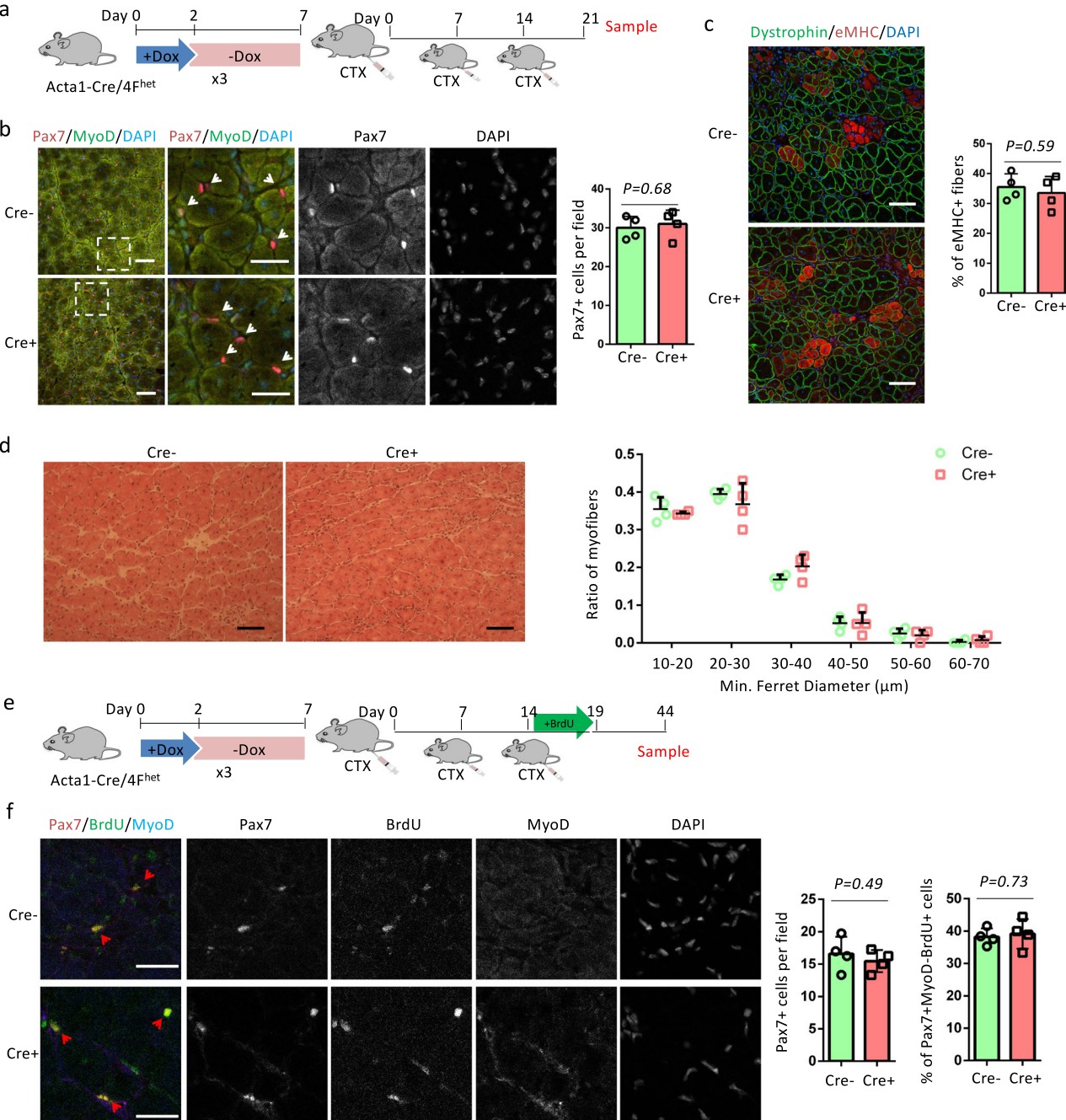

**Fig. 5 OSKM induction does not change the self-renewal of SCs. a** Schematic representation of the experimental design for 7 days after triple CTX injury. **b** Immunostaining of Pax7 and MyoD, and quantification of Pax7 cells in TA muscle sections. Scale bars = 50 μm. Representative regions are shown at higher magnification. Scale bars = 25 μm. White arrows indicate Pax7⁺ cells. **c** Immunostaining of embryonic myosin heavy chain (eMHC) and Dystrophin in TA muscle sections, and the quantification of the percentage of immature myofibers that express eMHC. Scale bars = 50 μm. **d** H&E staining of TA muscle sections and myofiber size distributions in TA muscle sections. Scale bars = 100 μm. **e** Schematic representation of the experimental design for 30 days after triple CTX injury. **f** Immunostaining of Pax7, BrdU, and MyoD, and quantification of Pax7 cells and percentage of BrdU⁺ SCs in TA muscle sections. Scale bars = 25 μm. Red Arrows indicate Pax7⁺MyoD⁻BrdU⁺SCs. Error bars represent mean + SD of four mice. A two-sided unpaired Student's *t*-test was performed.

injection after cyclic OSKM induction (Supplementary Fig. 8 g). Cre⁺ muscles had fewer eMHC⁺ myofibers compared with Cre⁻ muscles (Supplementary Fig. 8 h, i). In addition, Cre⁺ muscles had more SCs than Cre⁻ muscles (Supplementary Fig. 8j, k). Moreover, myofibers in Cre⁺ muscles were larger than those observed in Cre⁻ muscle, with more myofibers that were 30–40 μm and fewer myofibers that were 10–20 μm in diameter (Supplementary Fig. 8 l). Together, these results indicate that

myofiber-specific OSKM induction promotes SC activation and proliferation and accelerates muscle regeneration independent of aging.

We previously showed that systemic OSKM induction could promote muscle regeneration in 12-month-old systemic 4F mice[8]. We decided to compare the changes of SCs before CTX injection in systemic 4F mice and Acta-Cre/4F^het mice. We did immunostaining of Pax7 and MyoD in TA muscles of 12–15-month-old

systemic 4F mice, which were injected with three cycles of Dox or PBS (as control) according to our previous study[8] (Supplementary Fig. 9a). We found that the percentage of activated (MyoD[+]) SCs was increased by 1.7-fold (Supplementary Fig. 9b, c). We did not observe a significant difference in the number of Pax7[+] cells as seen in Acta-Cre/4F[het] mice (Supplementary Fig. 9b, d). To analyze the differences, we first compared the induction level of OSKM. The expression level of OSKM in skeletal muscles of systemic 4F mice was 5–10 times lower than that in Acta-Cre/4F[het] mice which were administrated with doxycycline in water (Supplementary Fig. 9e, f). In addition, p21 was induced in the SCs of systemic 4F muscles but not in the SCs of Acta-Cre/4F[het] mice (Supplementary Fig. 9 g). The induction of p21 could negatively regulate the $G_1$ to S progression of satellite cells[19]. Our results provide an additional explanation for the improved muscle regeneration in systemic 4F mice besides SC rejuvenation.

**OSKM in myofibers inhibits Wnt4 expression through p21**. We next focused on understanding the mechanisms by which OSKM induction in myofibers activates SCs. We analyzed the levels of expression of genes related to the SC microenvironment. RNA-seq analysis revealed that 2.5 and 8.5 days of Dox treatment affected the expression of 13 niche-related genes in Cre[+] muscles (Fig. 6a). Expression of four of these genes (*Itga7*, *Sdc4*, *Wnt2b*, and *Wnt4*) was altered on average over two-fold, and among these, *Wnt4* was the only one that was downregulated (Fig. 6a). Integrin subunit alpha 7 (Itga7) is a laminin-binding protein that connects myofibers to the basal lamina to maintain sarcolemma integrity[20]. It is unclear what role Wnt family member 2b (Wnt2b) plays in skeletal muscle, but it may not be important for myofibers since it has a low expression level (fpkm < 0.1). Syndecan-4 (Sdc4) is expressed in SCs and is required for SC activation[21,22], but a role for Sdc4 in myofibers remains elusive. Wnt family member 4 (Wnt4) is a recently identified paracrine factor that is secreted from myofibers[23]. Loss of *Wnt4* in myofibers induces the activation of SCs and accelerates muscle regeneration[23]. In mouse keratinocytes, p21 inhibits *Wnt4* expression by binding to the TATA-proximal region of the *Wnt4* promoter in association with E2F transcription factor 1 (E2F-1)[24], and levels of *p21* expression were negatively correlated with *Wnt4* expression in our RNA-seq analysis (Fig. 6b). We verified expression changes for *p21* and *Wnt4* in TA muscles of Acta1-Cre/4F[het] mice. Levels of *p21* and *Wnt4* expression were upregulated and downregulated in Cre[+] muscles, respectively, following Dox treatment, and returned to control levels (i.e., levels seen in Cre[−] samples) following Dox withdrawal (Fig. 6c).

Since Wnt4 is secreted from myofibers, we assessed *Wnt4* expression in single myofibers isolated from EDL muscles of Acta1-Cre/4F[het] mice. Isolated myofibers were cultured for 2 days in Dox (1 µg/ml), which induced the expression of OSKM in Cre[+] myofibers (Fig. 6d). The expression of *p21* and *Wnt4* were significantly upregulated and downregulated, respectively, in Cre[+] myofibers compared to Cre[−] myofibers (Fig. 6e). It is known that OSKM induction activates the p53–p21 pathway[25]. Consistent with this, in vivo levels of p53 and p21 protein were increased in Cre[+] muscles isolated from Acta1-Cre/4F[het] mice after 2.5 days of Dox treatment (Fig. 6f). To further validate that p21 downregulates *Wnt4* expression, we overexpressed p21 in TA muscles of wild-type mice through the electroporation of exogenous plasmids (Fig. 6g). Electroporation of the CAG-3xFLAG-p21 plasmid increased *p21* expression 60-fold, and downregulated *Wnt4* levels 3-fold, compared with the control plasmid (CAG-Td) (Fig. 6g).

We next performed chromatin immunoprecipitation (ChIP) to determine whether p21 could bind to the TATA-proximal region

of the *Wnt4* promoter in skeletal muscles. We overexpressed FLAG-tagged p21 or Tdtomato (control) in TA muscles (Fig. 6g), and then used a FLAG antibody or mouse IgG for ChIP analysis. The ChIP assay showed a 3-fold enrichment of the TATA-proximal region in muscles with p21 overexpression (Fig. 6h). In contrast, there was no enrichment of the TATA-proximal region in control muscles (Fig. 6h). We further designed a luciferase reporter (P[Wnt4]−Pmin-Luc), which contains the TATA-proximal region upstream of a minimal CMV promoter (Pmin), to determine whether the TATA-proximal region was responsible for the p21-mediated *Wnt4* repression (Fig. 6i). Notably, the TATA-proximal region may bind with E2F-1[24], leading to the increased activity of Pmin (Fig. 6i). Whereas p21 was unable to affect the control reporter (Pmin-Luc), which lacks the TATA-proximal region, p21 inhibited the P[Wnt4]−Pmin-Luc reporter by 2-fold (Fig. 6i). These results prove that OSKM induction downregulated the expression of *Wnt4* through p21.

**Wnt4 diminishes OSKM-induced SC changes**. The repression of Wnt4 in myofibers has been suggested to activate MyoD and Yap to elicit the activation of SCs[23] (Fig. 6j). As shown above, we observed MyoD activation in Cre[+] muscles (Fig. 3f, j). We next examined the levels of active Yap in SCs. As expected, levels of active Yap were higher in Pax7[+] cells from Cre[+] myofibers than in Pax7[+] cells from Cre[−] myofibers (Supplementary Fig. 10a, b). In addition, the percentage of Pax7[+]MyoD[+] active Yap[+] cells was 3-fold higher in Cre[+] myofibers than in Cre[−] myofibers (Supplementary Fig. 10b). We next investigated whether recombinant Wnt4 (rWnt4) could diminish the changes seen in SCs in Cre[+] myofibers. We isolated single myofibers from EDL muscles of Acta1-Cre/4F[het] mice treated with Dox for 2 days and cultured in plates with Dox and rWnt4 (100 ng/ml) or PBS (control treatment) for 8 or 48 h (Supplementary Fig. 10c). Whereas rWnt4 did not change the expression of active Yap and MyoD in Pax7[+] cells from Cre[−] myofibers, rWnt4 significantly decreased the expression of MyoD and abolished the upregulation of active Yap in Pax7[+] cells from Cre[+] myofibers 8 h after treatment (Supplementary Fig. 10d–f). More Pax7[+] cells per cluster were found in Cre[+] myofibers compared to Cre[−] myofibers 48 h after treatment, which was abrogated by rWnt4 treatment (Supplementary Fig. 10g). These results indicate that the repression of Wnt4 was responsible for OSKM induction-induced SCs activation and proliferation by activating MyoD and Yap.

**Cyclic *Wnt4* knockdown facilitates muscle regeneration**. Since *Wnt4* was an effective downstream target of OSKM, we further examined muscle regeneration after cyclic *Wnt4* knockdown. We chose CasRx to target *Wnt4* mRNA, as CasRx-mediated knockdown has been shown to be efficient and specific[26]. Moreover, a Tet-on system could be utilized to control the expression of CasRx for the purpose of cyclic *Wnt4* knockdown. We first tested the knockdown efficiency of three gRNAs with top ranks (minimum off-targets) in CHOPCHOP[27]. We generated a reporter vector, in which a part of the *Wnt4* cDNA containing the gRNA targets was inserted between the stop codon of *mCherry* and *polyA* (Supplementary Fig. 11a). With an efficient gRNA, CasRx is supposed to remove *polyA* from *mCherry*, expediting the degradation of mCherry mRNA to diminish the mCherry signal. Our results showed that all three gRNAs could efficiently diminish the mCherry signal (Supplementary Fig. 11a). Next, we quantified the knockdown efficiency of the gRNAs. Plasmids containing *CasRx*, *Wnt4* cDNA and a gRNA were co-transfected into HEK293 cells. Two days after transfection, the expression of *Wnt4* was quantified by 2 pairs of primers (Supplementary Fig. 11b). gRNA1 decreased the expression of *Wnt4* by 85%, and

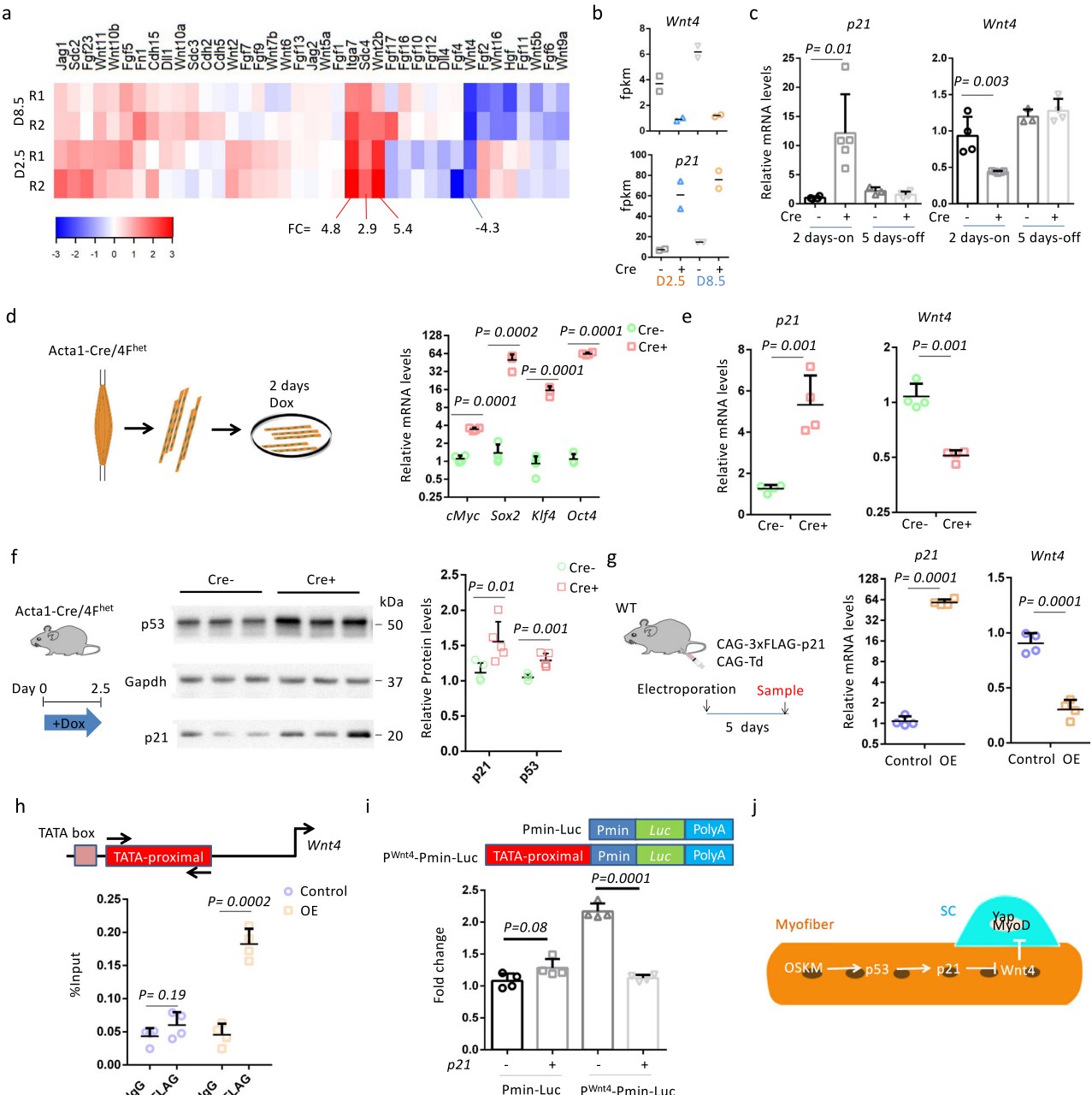

**Fig. 6 OSKM induction represses the expression of Wnt4 through p21. a** Heatmap shows the fold change of SCs niche-related genes in Cre⁺ EDL muscles compared to Cre⁻ EDL muscles. **b** The fpkm of *Wnt4* and *p21* in Cre⁻ and Cre⁺ EDL muscles. *n* = 2 independent biological samples. **c** Relative mRNA levels of *Wnt4* and *p21* in TA muscles after 2-days Dox treatment (2 days-on) and after 5-days Dox withdrawal (5 days-off). *n* = 4 Cre- mice for 2 days-on and 3 Cre− mice for 5 days-off. *n* = 5 Cre+ mice for 2 days-on and Cre+ mice for 5 days-off. Error bars represent mean + SD. **d** Relative mRNA levels of OSKM in single myofibers after 2-days Dox treatment. Error bars represent mean + SD of four independent biological samples. **e** Relative mRNA levels of *p21* and *Wnt4* in single myofibers after 2-days Dox treatment. Error bars represent mean + SD of four independent biological samples. **f** Western blots detection and quantification of the protein levels of p53 and p21 in TA muscles of Cre⁻ and Cre⁺ mice. Gapdh is the internal control. Error bars represent mean + SD of Cre⁻ 6 mice and 5 Cre⁺ mice. **g** Relative mRNA levels of *p21* and *Wnt4* in TA muscles with CAG-3xFLAG-p21 or CAG-Td plasmids. Error bars represent mean + SD of four mice. **h** Enrichments of the TATA-proximal region of the *Wnt4* promoter by 3xFLAG-p21 as determined by ChIP assay using an antibody against FLAG. Error bars represent mean + SD of four independent biological samples. **i** Luciferase assay with a reporter containing the TATA-proximal region upstream of the minimal CMV promoter (Pmin). Error bars represent mean + SD of four independent biological samples. **j** Hypothetical model showing how OSKM regulates Wnt4. A two-sided unpaired Student's *t*-test was performed.

both gRNA2 and gRNA3 decreased the expression of *Wnt4* by around 75% compared with mock gRNA (Supplementary Fig. 11b). We chose gRNA1 to generate AAV (serotype 9) containing CasRx driven by a TetO promoter for in vivo delivery of the CasRx system (Fig. 7a). To test the in vivo *Wnt4* knockdown, the left and right TA muscles of R26-M2rtTA mice were injected

with 2 × 10¹¹ genome copies (GC) AAV-gRNA1-CasRx and AAV-CasRx (as control), respectively. Six weeks after AAV injection, we verified that *Wnt4* expression was decreased by 65% in left TA muscles compared to right TA muscles following Dox treatment, and returned to control levels following Dox withdrawal (Fig. 7b).

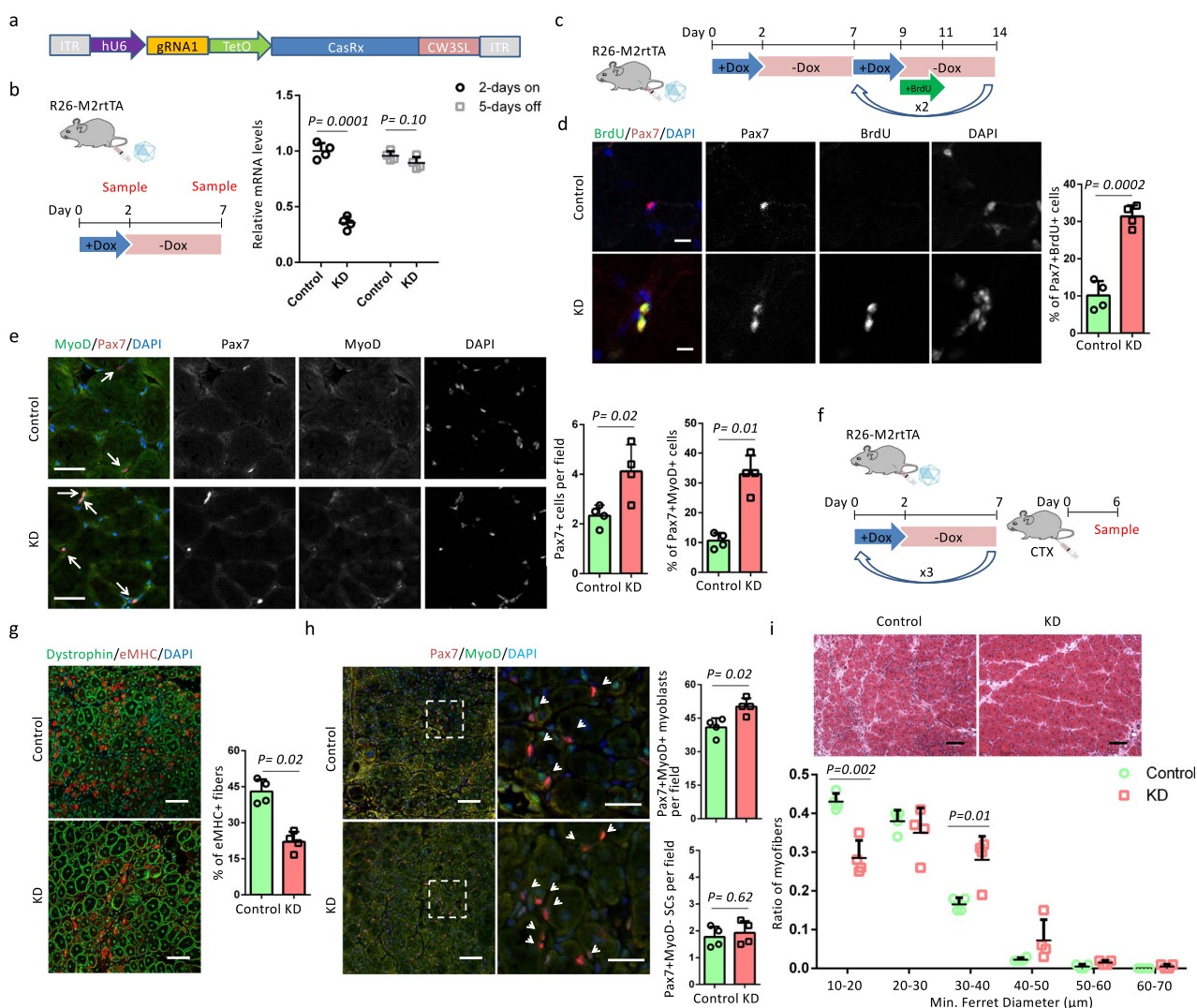

**Fig. 7 CasRx-mediated cyclic Wnt4 knockdown accelerates muscle regeneration. a** AAV-gRNA1-CasRx vector. **b** Relative mRNA levels of *Wnt4* in AAV-gRNA1-CasRx (KD)- or AAV-CasRx (Control)-treated TA muscles after 2-days Dox treatment and after 5-days Dox withdrawal. **c** Schematic representation of the experimental design. **d** Immunostaining of Pax7 and BrdU and quantification of the percentage of Pax7+BrdU+ cells in TA muscle sections. Scale bars = 10 μm. **e** Immunostaining of Pax7 and MyoD and quantification of Pax7+ cells per field and the percentage of Pax7+MyoD+ cells in TA muscle sections. Pax7+ cells are indicated by arrows. Scale bars = 50 μm. **f** Schematic representation of the experimental design. **g** Immunostaining of embryonic myosin heavy chain (eMHC) and Dystrophin in TA muscle sections, and the quantification of the percentage of immature myofibers that express eMHC. Scale bars = 50 μm. **h** Immunostaining of Pax7 and MyoD, and quantification of Pax7+MyoD− SCs and Pax7+MyoD+myoblasts in TA muscle sections. Scale bars = 50 μm. Representative regions are shown at higher magnification. Scale bars = 25 μm. Arrows indicate Pax7+ cells. **i** H&E staining of TA muscle sections and myofiber size distributions in TA muscle sections. Scale bars = 100 μm. Error bars represent mean + SD of four mice. A two-sided unpaired Student's *t*-test was performed.

Next, we investigated the status of SCs after cyclic *Wnt4* knockdown. Six weeks after AAV injection, R26-M2rtTA mice underwent three cycles of Dox. After the second and third rounds of Dox administration, BrdU was added to the drinking water for 2 days to label cycling SCs (Fig. 7c). Co-staining of BrdU and Pax7 showed that left muscles (with AAV-gRNA1-CasRx, KD) had a 3-fold higher percentage of Pax7+BrdU+cells than right (with AAV-CasRx, control) (Fig. 7d). In addition, KD muscles had 2-fold more Pax7+ cells and 3-fold more activated (MyoD+) SCs than control muscles (Fig. 7e). We then studied muscle regeneration capacity 6 days post CTX injection after cyclic *Wnt4* knockdown (Fig. 7f). KD muscles had 1-fold less eMHC+ myofibers compared with control muscles (Fig. 7g). In addition, KD muscles had more Pax7+MyoD+ myoblasts than control muscles (Fig. 7h). In contrast, the number of Pax7+MyoD− SCs was comparable in control and KD samples (Fig. 2i), indicating

Cre+ samples had an expansion of myoblasts without changing the SCs number. Moreover, myofibers in KD muscles were larger than those observed in control, with more myofibers that were 30–40 μm and less that were 10–20 μm in diameter (Fig. 7i). Together, these results indicate that cyclic *Wnt4* knockdown recapitulated OSKM induction to induce SC activation and myoblasts expansion to promote muscle regeneration.

## Discussion

Our work demonstrates that partial reprogramming via OSKM can remodel a stem cell niche to induce stem cell activation and proliferation and accelerate muscle regeneration. Specifically, expressing OSKM in myofibers induced p53–p21 signaling, which subsequently inhibited *Wnt4* transcription. The downregulation of *Wnt4* contributed to the activation of MyoD and Yap in SCs,

resulting in their proliferation. Although SC activation and proliferation were demonstrated to be directly regulated by myofibers, it might be indirectly affected by the extracellular matrix (ECM) or the other types of cells in the SC niche[29,30], as myofibers could secrete exosome to regulate other cells[28]. Reprogramming factors have been shown to promote axon regeneration through intrinsic rejuvenation of retinal ganglion cells[10]. In contrast, our results suggest an aging-independent effect of reprogramming factors on muscle regeneration through remodeling of the extrinsic niche but not by regulating the intrinsic modulators of SCs.

Intrinsic modulators control SC-initiated postnatal myogenesis[29]. The intrinsic regulatory machinery is efficient in young SCs, therefore, further improvement of SC quality is limited. In addition, SC-specific OSKM induction did not change the quantity of SCs or myoblasts and thus no difference in muscle regeneration was observed. Our results suggest that modifying the stem cell niche could be a definitive intervention to boost young tissue regeneration. Notably, the timing of this targeting is an important factor. As injury is a strong signal to trigger SC proliferation, the effect of OSKM on SC activation and proliferation was negligible after injury, and thus post-injury induction of OSKM did not accelerate muscle regeneration.

We compared the changes in SCs in aging Acta-Cre/4F[het] and systemic 4F mice. Both myofiber-specific and systemic OSKM induction increased the activation of SCs. SCs activation is required for muscle regeneration, but the activation kinetics is delayed in aged SCs[30]. The increase in activated SCs accelerated muscle regeneration, which may partly explain our previous observation that muscle regeneration was promoted in aging systemic 4F mice[8]. SC activation could be a result of OSKM induction in myofibers. We cannot exclude the possibility that other cell types (macrophages, endothelial cells, fibroblasts, etc.) also contribute to the activation of SCs[31–33]. Systemic OSKM induction has been shown to mitigate the decline of Pax7[+] cells in progeria mice after 6 weeks of cyclic induction of OSKM[8]. In contrast, systemic OSKM induction did not change the number of Pax7[+] cells in naturally aging mice after 3 weeks of cyclic induction of OSKM. The aging process is much slower in WT mice compared to progeria mice and longer time may be required to observe the effect of OSKM on SC number in naturally aging mice. Whereas myofiber-specific OSKM induction increased SC proliferation, OSKM induction in SCs may prevent the proliferation of SCs in systemic 4F mice.

Self-renewal of SCs is maintained by the SC niche, including, but not limited to ECM components and myofibers[34,35]. Remodeling the niche may impair the self-renewal of SCs, which is crucial for long-term muscle maintenance. Deletion of collagen VI or laminin-α1 (ECM components) or myofiber-specific deletion of O-fucosyltransferase 1 gene (Pofut1) impairs the self-renewal of SCs[36–38], leading to the shrinkage of the SC pool after multiple muscle injuries. Interestingly, diminishment of the SC pool is not observed when SCs-SC niche molecular interactions are interrupted in muscles lacking Wnt4, N-, and M-cadherins or syndecan-3[23,39,40]. In our model, the myofiber-specific OSKM induction did not change the number and state of SCs after multiple muscle injuries. One possible explanation is that the SC pool may establish a homeostatic mechanism similar to the one that is observed in muscles lacking Wnt4[23]. Another possibility is that SC homeostasis was not changed, as the SC niche was only temporarily modified. The two possibilities may be distinguished by investigating the heterogeneity of SCs in future studies.

OSKM induction triggers complex cellular changes[1–3]. The identification of their effective downstream targets would restrict unwanted outcomes and benefit the development of new approaches to facilitate tissue regeneration. We identified Wnt4

as a downstream target of OSKM and found that cyclic Wnt4 knockdown could facilitate SC activation and muscle regeneration. Interestingly, the effect of cyclic Wnt4 knockdown on muscle regeneration is similar to permanent Wnt4 knockdown[23]. From a translational point of view, myofibers are easily accessible by viral vectors, mRNAs or nanoparticles, thus making it feasible to devise safe strategies to introduce the CasRx-mediated Wnt4 knockdown system into myofibers to accelerate muscle regeneration and treat muscle degeneration-related diseases.

## Methods

**Mice.** Acta1-Cre[+] (Stock#006139)[41], Pax7[creER/+] (Stock#017763)[33], Col1a1-tetO-4F[homo] (Stock#011004)[42], systemic 4F mice (Stock#011001), ROSA26-LSL-rtTA[homo] (Stock#005670)[43], R26-M2rtTA (Stock#006965)[44], and Ai14[homo] (Stock#007908) mice were from Jackson Laboratory. Col1a1-tetO-4F[homo] mice were mated with ROSA26-LSL-rtTA[homo] mice to generate Col1a1-tetO-4F[het]ROSA26-LSL-rtTA[het] (het/het). Het/het mice were mated to generate Col1a1-tetO-4F[homo]ROSA26-LSL-rtTA[homo] (4F[homo]) mice. Acta1-Cre[+] and Pax7[creER/+] mice were mated with 4F[homo] mice, respectively, to generate Acta1-Cre/4F[het] (representing both Acta1-Cre[+]/4F[het] and Acta1-Cre[−]/4F[het]) and Pax7[creER]/4F[het] (Pax7[creER/+]/4F[het] and Pax7[+/+]/4F[het]) mice. Acta1-Cre[+]/4F[het] and Ai14[homo] mice were mated to generate Acta1-Cre[+]/4F[het]/Ai14[het]. We used 6-weeks to 4-month-old as young mice and 12–15-month-old as aging mice, both male and female mice for this study. All animal procedures were performed according to protocols approved by the IACUC and Animal Resources Department of the Salk Institute for Biological Studies.

**In vivo treatment.** Tamoxifen (Sigma–Aldrich) was prepared in corn oil at a concentration of 10 mg ml[−1], and Pax7[creER]/4F[het] mice were injected intraperitoneally with 2 mg TAM per day per 20 g body weight for 5 days to induce Cre-mediated deletion. Dox (RPI) was administered in drinking water for 2 or 2.5 days at a concentration of 1 mg ml[−1] to induce OSKM expression. BrdU (ApexBio) was administered in drinking water for 2 days after Dox treatment at a concentration of 1 mg ml[−1] with 5% glucose to label cycling cells. CTX (Latoxan S.A.S.) was injected into the left TA muscle to induce muscle regeneration at a concentration of 10 μM and a volume of 40 μl after mice were anesthetized using a ketamine (100 mg kg[−1])-xylazine (10 mg kg[−1]) cocktail. In vivo muscle electroporation was following the established protocol[45]. Wild-type mice were anesthetized with intraperitoneal injection of the ketamine-xylazine cocktail. Plasmid DNA (30 μg) was injected into the muscle using a 29-gauge insulin syringe. One minute following plasmid DNA injection, a pair of electrodes was inserted into the TA muscle to a depth of 5 mm to encompass the DNA injection site. Muscle was electroporated using an Electro Square Porator T820 (BTX Harvard Apparatus). Electrical stimulation was delivered in three pulses at 100 V for 50 ms followed by three more pulses of the opposite polarity.

**Single myofiber isolation.** EDL muscles were removed carefully and digested with 2 mg ml[−1] collagenase type 1 (Worthington) in Dulbecco's Modified Eagle's Medium (DMEM, Gibco) for 45 min[−1] h at 37 °C. Digestion was stopped by carefully transferring EDL muscles to a horse serum-coated Petri dish (60-mm) with DMEM. Myofibers were released by gently flushing muscles with a large bore glass pipette. Released single myofiber was either fixed immediately with 4% PFA or transferred and cultured in a horse serum-coated Petri dish (60-mm) in DMEM supplemented with 20% fetal bovine serum (FBS, Gibco), 4 ng ml[−1] basic fibroblast growth factor (bFgf, Peprotech), and 1% penicillin–streptomycin (P/S, Gibco) at 37 °C for indicated days following the protocol by Yue and colleagues[46]. Dox (RPI) was administered in culture medium at a concentration of 1 μg ml[−1] to induce OSKM expression. Recombinant Wnt4 (R&D systems) was administered in culture medium at a concentration of 100 ng ml[−1] according to a previous study[23].

**Primary myoblast isolation, culture, and differentiation.** Primary myoblasts were isolated from 5-week-old Pax7[creER]/4F[het] female mice. The hind limb skeletal muscles were minced and digested in type I collagenase (Worthington) and dispase B (Sigma) mixture. The digestion was stopped with F-10 Ham's medium containing 20% FBS, and the cells were filtered from debris, centrifuged and cultured in growth media (F-10 Ham's media supplemented with 20% FBS, 4 ng ml[−1] bFgf and 1% P/S) on uncoated dishes for three days when 5 ml growth media were added each day. Then the supernatant was collected, centrifuged and trypsinized with 0.25% trypsin. After washing off the trypsin, primary myoblasts were seeded on collagen-coated dishes, and the growth medium was changed every two days. Myoblasts were treated with 1 μg ml[−1] Dox in growth media.

**Immunostaining and H&E staining.** Muscle samples were collected and processed for immunostaining and H&E staining following the protocol described by Wang et al.[47]. Muscle slides and single myofibers were fixed with 4% paraformaldehyde (PFA) for 5 min. Then samples were washed with PBS for at least three times

followed by incubation of 100 mM Glycine for 15 min. After washing with PBS, samples were blocked with blocking buffer (5% goat serum, 2% BSA, 0.2% triton X-100, and 0.1% sodium azide in PBS) for at least 30 min. Primary antibodies were diluted with blocking buffer. Anti-Pax7 (DSHB, Pax7-c), anti-MyoD (SCBT, sc-377460, G-1), anti-BrdU (Abcam, ab6326, [BU1/75 (ICR1)]), anti-Sox2 (Cell Signaling, #2748), anti-Oct-3/4 (SCBT, sc-5279, C-10), anti-GFP (SCBT, sc-9996, B-2), and anti-active Yap (Abcam, ab205270, EPR19812) were diluted at a ratio 1:200. Anti-eMHC (DSHB, F1.652) was diluted at a ratio 1:80. Dystrophin (Abcam, ab15277) antibody was diluted at a ratio 1:500. Blocking buffer was removed from the sections and diluted primary antibodies were added on sections overnight at 4 °C. After washing with PBS, the samples were incubated with respective secondary antibodies and DAPI for 1 h at room temperature. Goat anti-Mouse IgG1 (Alexa Fluor 568, A-21124), Goat anti-Mouse IgG2b (Alexa Fluor 488, A-21141), Goat anti-Mouse IgG2b (Alexa Fluor 647, A-21242), Donkey anti-Rabbit IgG (H + L) (Alexa Fluor 488, A-21206), and Donkey anti-Rat IgG (H + L) (Alexa Fluor 488, A-21208) were from Thermo Fisher Scientific. All the secondary antibodies were diluted at a ratio 1:500. For BrdU staining, the muscle slides were pretreated with DNase I solution (25 µl DNase I diluted in 1 ml RDD buffer) for 1 h at 37 °C followed by the PFA treatment. The DNase I and RDD buffer were provided in the RNase-free DNase set (Qiagen). Fluorescent images were captured using a Zeiss LSM 710 Laser Scanning Confocal Microscope. Images were quantified by using the ImageJ software. H&E staining was exactly following the protocol described by Wang et al.[47]. H&E images were captured using an Olympus microscope IX51.

### RNA extraction and real-time qPCR.
Total RNA of muscles, myofibers and myoblasts were extracted using Trizol Reagent (Ambion). The muscles and myofibers were homogenized by using EpiShear Probe Sonicator. RNA was treated with RNase-free DNase I to remove genomic DNA. The purity and concentration of total RNA were measured by Synergy H1 (BioTek). cDNA was generated by reverse transcription using Maxima H Minus Reverse Transcriptase (ThermoFisher Scientific). For mature miRNA reverse transcription, multiple adenosine nucleotides were first added to the 3' end of total RNA with Escherichia coli DNA polymerase (NEB), and cDNA was then synthesized with a PolyT primer including an adaptor sequence[48]. SsoAdvanced Universal SYBR Green Supermix (BioRed) was used to carry out the qPCR analysis in CFX 384 Real-time System (BioRed). The expression levels of respective genes were normalized to the housekeeping gene GADPH and miRNA to sno202[48]. Primers sequences were listed in Supplementary Table 1.

### RNA-seq analysis.
Raw reads were aligned to the mm10 genome using STAR [v2.5.3a][49] using default parameters. Then the number of reads uniquely aligned to RefSeq exons were quantified by HOMER [v4.9.1][50]. Samples without OSKM overexpression were assessed as outliers and removed from downstream analysis. Principle Component Analysis (PCA) was performed on the top 500 most variable genes using the log-transformed fragments per kilobase per million mapped (FPKM) values by R function "prcomp". A pseudo-count of 5 was added to each value before log-transformation to avoid infinite values. R package "ggplot2" (https://cran.r-project.org/web/packages/ggplot2/index.html) was used to generate the PCA plot. Differential expression (DE) analysis was performed on the raw reads using DESeq2 [v1.22.2][51] for pair-wise comparisons between the control and treatment samples at each time point using default normalization and dispersion estimation. Genes with adjusted $p$-value < 0.05 and absolute log fold-change >0.5 were identified as significantly differentially expressed genes between two conditions. A union set of 3858 DE genes of all four pair-wise comparisons were used for the pattern analysis by WGCNA [v1.67, https://bmcbioinformatics.biomedcentral.com/articles/10.1186/1471-2105-9-559] on the z-scaled log-transformed FPKM values. A soft power of 30 was used to detect modules and a dissimilarity score of 0.25 was used to merge similar modules. Heatmap was generated on 1000 downsampled genes using a random seed of 12345 and plotted by R package gplots [https://cran.r-project.org/web/packages/gplots/index.html]. Boxplot showed the distribution of the mean of the z-scaled gene expression across replicates. Significantly over-represented Gene Ontology (GO) terms were identified by R package WebGestaltR [v0.3.1, https://academic.oup.com/nar/article/45/W1/W130/3791209] using the gene list of each module as input and total RefSeq genes as background. Terms with adjusted $p$-value < 0.05 were identified as significant. Terms with adjusted $p$-value < 1e-20 were shown as 1e-20 for the bar plots.

### Protein extraction and western blot analysis.
Muscle samples were washed with PBS and homogenized with radioimmune precipitation assay buffer (50 mM Tris-HCl (pH 8.0), 150 mM NaCl, 1% NP-40, 0.5% sodium deoxycholate, and 0.1% SDS). Proteins (100 ug) were separated by 4–20% precast polyacrylamide gel (Biored), electrotransferred onto PVDF membrane (Millipore), and incubated with specific primary antibodies. Anti-p53 (Cell Signaling, #2524 S) was diluted at a ratio of 1:100, and anti-p21 (Abcam, ab109520) and anti-Gapdh (Cell Signaling, #2188 S) were diluted at a ratio of 1:1000 in 5% w/v nonfat dry milk. Secondary antibodies, the Anti-mouse IgG, HRP-linked (Cell Signaling, #7076) and anti-Rabbit IgG, HRP-linked (Cell Signaling, #7074) were diluted in Tris-buffered saline with 0.1% Tween 20 detergent (TBST) at a ratio of 1: 2500. Immunodetection was performed using SuperSignal West Pico PLUS Chemiluminescent Substrate (Thermo Scientific).

### Chromatin immunoprecipitation.
The chromatin immunoprecipitation (ChIP) assay was carried out by using the Zymo-Spin ChIP kit (D5209, Zymo Research) according to manufacturer's protocol. Specifically, 20 mg TA muscles transfected with CAG-Td or CAG-3xFLAG-p21 were isolated and homogenized in PBS with a Dounce homogenizer. After the steps of formaldehyde cross-linking and nuclei preparation, we did the chromatin shearing by using a Covaris M220 Focused-ultrasonicator with indicated parameters, including Peak Incident Power (75 W), Duty factor (15%), Cycles per burst (200), and Treatment time (1500 s). Then, 10% of sheared chromatin was set aside as a DNA input. The remaining sheared chromatin was equally separated for ChIP reaction by using 1 µg anti-FLAG antibody (Sigma, F1804, M2) or 1 µg mouse IgG (SCBT, sc-2025). The Wnt4 promoter TATA-proximal region was amplified by the primer pair listed in Supplementary Table 1.

### Luciferase assay.
Transient transfections were performed with Lipofectamine3000 (ThermoFisher Scientific) according to manufacturer's instructions. N2a cells were grown in Dulbecco's modified Eagle's media (DMEM) supplemented with 10% FBS. Briefly, 200 ng of reporter Pmin-$Luc$ or $P^{wnt4}$–Pmin-$Luc$, 10 ng Renilla, and 200 ng of CAG-3xFLAG-p21 or CAG-Td were transfected to N2a cells in 12-well plates. After 48 h, cells were harvested and analyzed with the Dual-Luciferase Reporter Assay System (Promega). The total amount of DNA added in each transfection was kept constant by referring to the Renilla signal. The Pmin-$Luc$ was generated by Liao et al.[52]. $P^{wnt4}$–Pmin-Luc reporter was generated by cloning the TATA-proximal region of Wnt4 promoter into Pmin-Luc reporter[24].

### CasRx-mediated Wnt4 knockdown.
The CHOPCHOP (https://chopchop.cbu.uib.no/) was adapted to choose gRNA targets of Wnt4. The top three gRNA targets (gRNAT1: CTTCCAGTGGTCAGGATGCTCGGACAAC; gRNAT2: AGCCACGACGCGTAGGCTCCTCCCGGGC; gRNAT3: TAAAGGAGAAGTTTGACGGTGCCACGG) were cloned into the gRNA backbone (Addgene#109053). We generated a reporter vector (CAG-mCherry-Wnt4-pA) by replacing the WPRE element of AAVS1-Pur-CAG-mCherry vector (Addgene#80946) with a part of Wnt4 cDNA containing the gRNA targets. To detect the knockdown efficiency of each gRNA, 400 ng CAG-mCherry-Wnt4-pA, 400 ng EF1a-CasRx-2A-EGFP (Addgene#109049) and 200 ng each gRNA or mock gRNA (GCACTACCAGAGCTAACTCATTCGAGAA) were transfected into HEK293 cells. The mCherry signal was observed 2 days after transfection. In addition, the knockdown efficiency of the gRNAs was quantified by transfecting HEK293 cells with 400 ng EF1a-CasRx-2A-EGFP, 400 ng CAG-Wnt4, and 200 ng each gRNA or mock gRNA. Two days after transfection, the expression of Wnt4 was quantified with qPCR by two pairs of primers listed in Supplementary Table 1. Then, CasRx was subcloned and inserted into an AAV vector (AAV2 inverted terminal repeat vector) under the control of a TetO promoter to generate AAV-CasRx vector. AAV-gRNA1-CasRx vector was generated by inserting gRNA1 into AAV-CasRx vector. The recombinant AAV vectors were pseudo-typed with AAV9 capsid and the viral particles are generated following the procedures of the Gene Transfer Targeting and Therapeutics Core at the Salk Institute for Biological Studies.

### Statistical analysis.
The data presented were taken from distinct samples with mean and standard deviation (SD). $P$-values were calculated using two-tailed unpaired Student's t-test. All analyses were performed with Prism 7 software. $P$-values <0.05 were considered to be statistically significant.

### Reporting summary.
Further information on research design is available in the Nature Research Reporting Summary linked to this article.

## Data availability
The original data of RNA-seq are deposited to GEO dataset (GSE148911, https://www.ncbi.nlm.nih.gov/geo/query/acc.cgi?acc=GSE148911). All data supporting the findings of this study are available from the corresponding authors upon reasonable request. Source data are provided with this paper.

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

## Acknowledgements

We thank D. O'Keefe for critical reading of this manuscript and M. Schwarz for administrative support. We thank Tong Zhang from Salk Biophotonics core. We thank the Razavi Newman Integrative Genomics and Bioinformatics Core Facility of the Salk Institute with funding from NIH-NCI CCSG: P30 014195, and the Helmsley Trust. R.R.R. was partially supported by the Fundacion Ramon Areces. This work was supported by Asociación de Futbolistas Españoles (AFE), Fundacion Dr. Pedro Guillen, Universidad Católica San Antonio de Murcia (UCAM), The Moxie Foundation, Fundación MAPFRE and CIRM (GC1R-06673-B).

## Author contributions

C.W. and J.C.I.B. conceived the project, designed the experiments, and prepared the manuscript. C.W., R.R.R., P.M.R., Z.M., I.G.G., and H.K.L. performed the experiments and analyzed the data. L.S and L.H. performed RNA-seq analysis. C.W., R.R.R., and Y.X. performed image analysis. T.H., E.N., C.R.E., P.G., and P.R. provided key reagents and technical assistance.

## Competing interests

The authors declare no competing interests.
