## [Peer Review File · Nature Communications]

REVIEWER COMMENTS

Reviewer #1 (Remarks to the Author):

The present work by Wang et al investigates the impact of transient upregulation of the 4 Yamanaka factors, Oct4, Sox2, Klf4 and c-myc, in skeletal muscle myofibers. The authors have previously shown that transient upregulation of the 4 Yamanaka factors ameliorates age-associated phenotypes in mice, and in the current manuscript they explore the specific role of this intervention in myofibers. The authors take advantage of a Dox- inducible mouse model to transiently express the four factors in myofibers and show that this leads to significant changes in gene expression as well as an increase in the number of Pax7+ and MyoD+ cells as well as their proliferation in the tissue and on isolated myofibers. Upon tissue injury and regeneration, the authors show accelerated tissue repair, shown by an increase in myofiber size and a decrease in the percentages of regenerating embryonic myosin heavy chain myofibers. Through RNAseq analyses, the authors show a decrease in the expression of Wnt4, previously shown to promote SC quiescence, and an increase in p21 and Yap. Finally, the authors show that when the Yamanaka factors are expressed directly in SC, this has no effect on SC numbers, activation status, or tissue repair. The authors conclude that transient expression of the Yamanaka factors in myofibers leads to upregulation of p21, which in turn downregulates Wnt4, leading to spontaneous SC activation and expansion. Developing novel approaches to improve tissue repair is relevant for the future development of interventions for ameliorating muscle diseases. While the findings are interesting, the study is preliminary and has limited cellular and mechanistic depth. Additional experiments are required in order to fully support the authors' interpretation.

Main points:

- 1- Expansion: While the authors interpret the observed increase in the number of Pax7+ cells and MyoD+ cells as expansion, it is unclear to this reviewer which stages of myogenic progression the treatment is actually expanding. Percentage of Pax7+/MyoD-, Pax7+/MyoD+ and Pax7-/MyoD+ cells should be assessed, in order to evaluate whether the treatment is indeed promoting expansion of SC (Pax7+MyoD-), myogenic progenitors (Pax7+MyoD+ cells) or myoblasts (Pax7-MyoD+).
- 2- Self-renewal: The authors show that the number of Pax7+ SC is increased in these mice, as well as their percentage of MyoD+, which is a marker of myogenic commitment. However, they do not provide any evidence that these cells are able to self-renew, or they just transition to the progenitor stage and deplete the SC pool. What is the number of Pax7+ SC at the end of the regenerative process, day 30? Are cells in these mice able to expand and re-enter quiescence for long-term tissue maintenance? Importantly, for assessment of self-renewal, serial injury experiments should be performed.
- 3- Expression Timing: The authors show that the number of SC is increased in these mice in the absence of tissue injury, however it is not evaluated what is their fate, i.e. do they spontaneously contribute to myofibers? Is there an increase in the number of centrally nucleated and embryonic myosin heavy chain+ myofibers in these mice in the absence of injury? In addition, as they emerge from quiescence before the injury is performed, the improved tissue repair might be the result of this early activation, compounded with the potential ongoing contribution to myofibers. What happens if the expression of the Yamanaka factors is induced concurrently with tissue injury?
- 4- Molecular mechanism: A causal relationship between p21, Wnt4 and the observed phenotype is not formally demonstrated. The authors should perform ChIP-qPCR to validate that p21 directly binds Wnt4a promoter in myofibers, as well as luciferase reporter assays with deletion mutants for the p21 binding region on the Wnt4 promoter, to demonstrate causality of the binding to the increased Wnt4a

transcription. In addition, that Wnt4a downregulation is mediating the observed phenotype should be established by rescue experiments, in which Wnt4a is upregulated or delivered in these mice. Finally, a discussion of the caveats should be included in the text, i.e. the observed phenotype could be either the result of a direct effect of myofibers on SC or an indirect effect through changes in other cell types in the tissue microenvironment. As several secreted factors (for example, Wnt4 could be upregulated also in other local cell types) or cell surface proteins could contribute to the phenotype.

Specific points:

1- In Figure 3 the authors show that in the absence of injury the number of SC are increased ~2 fold in the tissue. However, upon tissue injury their number is increased only ~30%, which might suggest that cells are not being expanded but rather rapidly differentiating. Can the authors reconcile these two results?

2- Quantification of western blot analysis in Figure 4F should be included to evaluate whether the difference in protein levels for p53 and p21 is statistically significant.

3- Figure S6c is unclear, it appears a repetition of experiments already shown in the manuscript, unclear if it is missing a panel here.

4- Statistical analysis, N, P value and statistical method used should be included in all figure legends (it is missing for Figure 1).

Reviewer #2 (Remarks to the Author):

In this manuscript, Wang, et al., reports that myofiber-specific, short-term (2 days), cyclic expression of Yamanaka reprogramming factors (OSKM) in young adult mice enables faster muscle regeneration upon injury due to increased number of satellite cells (SCs) via down-regulation of Wnt4, that maintain SCs quiescent.

This is a follow up study of their previous publication (Ocampo et al., 2016), in which they demonstrated that ubiquitous short-term cyclic expression of OSKM ameliorates hallmarks of ageing, including the increased regeneration capacity of muscle in 12 months old mice. The phenotype, reported as if it was caused by epigenetic remodelling during cellular reprogramming, was quite surprising and very interesting. I welcome this new work, in which they demonstrated that at least the enhanced muscle regeneration capacity by OSKM in young mice could be explained by the increased SC number preceding the injury, while I think there are several key questions to be addressed in association with the previous work as below.

1. What % of muscle fibres become GFP+ (successful Cre excision) and what % can express OSKM? It is not clear if all muscle fibres can induce OSKM expression. Even if it is not 100%, is it sufficient to explain the phenotype? Immunofluorescence for Oct4 (and perhaps p21), images of GFP expression upon dox administration would be informative.

2. Does the number of SCs increase even when OSKM are expressed systemically (i.e., in all cell types, including muscle fibre and SCs)? I expect SCs also up-regulate p21 expression in response to OSKM expression. Doesn't it inhibit SC proliferation? Or SCs do not up-regulate p21? Or the previous system did not have strong OSKM expression in SCs?

3. The observed phenotype in this work can be seen in old mice? i.e. the improved muscle

regeneration capacity in old mice in the previous paper can be explained by the increased number of SCs (caused by suppression of Wnt4)?

4. 50% reduction of Wnt4 (Fig 4c) is sufficient to explain the phenotype? Can addition of Wnt4 block SC proliferation in vitro, as shown in Figure S1G in <https://doi.org/10.1016/j.stem.2019.08.007> (Wnt4 from the Niche Controls the Mechano-Properties and Quiescent State of Muscle Stem Cells, Cell Stem Cell, 2019)?

5. Is it right to conclude that the increased regeneration capacity of muscle by OSKM expression in old mice (Ocampo, et al.,) was due to increased SC number, not due to epigenetic rejuvenation of the cells? While one might call 'increasing SCs' rejuvenation of muscle, it is not specific in old muscle. I believe it is important to clearly state that the increased regeneration capacity is a secondary effect of increased SC numbers, not due to magical rejuvenation of old cells (unless epigenetic rejuvenation is also important for old SCs and/or muscle to show this phenotype).

Minor points,

6. Did the authors sort GFP+ cells for the RNA-seq (if the excision efficiency is not high)?

7. Do you see increased SCs in the context of CAG-p21 electroporation (Fig 4g)?

Reviewer #3 (Remarks to the Author):

Previously the author's group reported that global over-expression of OSKM in mice is capable of inducing an expansion of muscle stem cells in older mice and improving muscle regeneration following injury. In this manuscript, the authors utilized muscle fiber-specific overexpression of OSKM in mice to examine whether the short-term OSKM-overexpressing niche has an effect on satellite cell (SC) function. As a result, OSKM-overexpressing induces SC activation and proliferation, increases in the number of SCs, and induces faster muscle regeneration. RNA-seq data showed increased expression of p21 and p53, and decreased expression of Wnt4 which recently showed an essential role in SC quiescence and a negative-regulator for SC differentiation, suggesting that short term OSKM-overexpressing may promote muscle regeneration by modifying the stem cell niche.

This is a follow-up manuscript after a recently published paper in which the global over-expression of OSKM can induce SC expansion and muscle regeneration in old mice. Overall, the mechanism for fiber-OSKM-mediated SC expansion is of interest, the molecular mechanism of this process is still preliminary.

Major issues:

1. OSKM-overexpressing can induce around 50% Wnt4 mRNA reduction. It is not so clear whether this reduction can induce SC expansion and enhanced muscle regeneration. The authors should address this question by Wnt4-overexpression, and Wnt4 gene knockdown and knockout mice.
2. Overall, muscle regeneration data is very preliminary. Please show the earlier phase as well as later time course after CTX injection. For example, the authors should examine day 2 or 3 for EdU/Pax7 staining and eMHC staining. In addition, day 14 and later stages for HE staining and fibrosis staining to see whether muscle regeneration is caught up at the later stages in OSKM-mice. Finally, it would be interesting to see whether OSKM-mice have better muscle function.
3. Since Wnt4 is important for SC-quiescence. It would be nice to try multiple CTX-mediated muscle injury in OSKM mice to see whether SC-quiescence is reduced by OSKM overexpression and muscle

regeneration capacity is diminished after multiple regeneration cycles.

Minor issues:

1. In Figure 3, it would be nice to show OSKM protein expression on muscle fibers.
2. In Figure 3, please show the ration of Pax7+, MyoD+ and Pax7+MyoD+ in single muscle fiber experiments.
3. In Figure 4i, please perform MyoD/YAP double staining and quantitative analysis on muscle fibers.

Response to reviewers' comments

We thank the reviewers for the positive evaluation of our work and appreciate their constructive suggestions, which helped to improve our study. We addressed the comments and performed additional experiments as suggested by the reviewers. Please see below our point-to-point responses.

Reviewer #1 (Remarks to the Author):

The present work by Wang et al investigates the impact of transient upregulation of the 4 Yamanaka factors, Oct4, Sox2, Klf4 and c-myc, in skeletal muscle myofibers. The authors have previously shown that transient upregulation of the 4 Yamanaka factors ameliorates age-associated phenotypes in mice, and in the current manuscript they explore the specific role of this intervention in myofibers. The authors take advantage of a Dox- inducible mouse model to transiently express the four factors in myofibers and show that this leads to significant changes in gene expression as well as an increase in the number of Pax7+ and MyoD+ cells as well as their proliferation in the tissue and on isolated myofibers. Upon tissue injury and regeneration, the authors show accelerated tissue repair, shown by an increase in myofiber size and a decrease in the percentages of regenerating embryonic myosin heavy chain myofibers. Through RNAseq analyses, the authors show a decrease in the expression of Wnt4, previously shown to promote SC quiescence, and an increase in p21 and Yap. Finally, the authors show that when the Yamanaka factors are expressed directly in SC, this has no effect on SC numbers, activation status, or tissue repair. The authors conclude that transient expression of the Yamanaka factors in myofibers leads to upregulation of p21, which in turn downregulates Wnt4, leading to spontaneous SC activation and expansion. Developing novel approaches to improve tissue repair is relevant for the future development of interventions for ameliorating muscle diseases. While the findings are interesting, the study is preliminary and has limited cellular and mechanistic depth. Additional experiments are required in order to fully support the authors' interpretation.

Main points:

1- Expansion: While the authors interpret the observed increase in the number of Pax7+ cells and MyoD+ cells as expansion, it is unclear to this reviewer which stages of myogenic progression the treatment is actually expanding. Percentage of Pax7+/MyoD-, Pax7+/MyoD+ and Pax7-/MyoD+ cells should be assessed, in order to evaluate whether the treatment is indeed promoting expansion of SC (Pax7+MyoD-), myogenic progenitors (Pax7+MyoD+ cells) or myoblasts (Pax7-MyoD+).

Response: We thank the reviewer for this suggestion. We quantified the percentage of Pax7⁺/MyoD⁻, Pax7⁺/MyoD⁺ and Pax7⁻/MyoD⁺ cells in muscle sections (**Fig. 3f**) and on myofibers (**Fig. 3j**).

2- Self-renewal: The authors show that the number of Pax7+ SC is increased in these mice, as well as their percentage of MyoD+, which is a marker of myogenic commitment. However, they do not provide any evidence that these cells are able to self-renew, or they just transition

to the progenitor stage and deplete the SC pool. What is the number of Pax7+ SC at the end of the regenerative process, day 30? Are cells in these mice able to expand and re-enter quiescence for long-term tissue maintenance? Importantly, for assessment of self-renewal, serial injury experiments should be performed.

Response: This is a valuable comment and we agree with the reviewer. To address whether OSKM induction affects the self-renewal of SCs, we performed additional experiments and the results are shown in two new figures (**Fig. 4 and Fig. 5**). First, we studied the state and number of SCs 14 and 30 days after CTX injection. As shown in figure 4h and 4i, the number of Pax7⁺ cells and the percentage of MyoD⁻Pax7⁺ cells were comparable in Cre⁺ and Cre⁻ muscles 30 days after CTX injection. Next, we investigated the state and number of SCs and muscle regeneration capacity after three times CTX injection (**Fig. 5**). Our results show that the number of Pax7⁺ cells were comparable in Cre⁺ and Cre⁻ muscles 7 and 30 days after three times CTX injection. Moreover, the muscle regeneration capacity was normal in Cre⁺ muscle after three times CTX injection. These results prove that the self-renewal of SCs was maintained after OSKM induction.

The self-renewal of SCs is affected by intrinsic regulators and extrinsic regulators. Our immunostaining data prove that OSKM was induced in the nuclei of myofibers and not in SCs (**Fig. 2b**), excluding the possibility of direct intrinsic changes of SCs. As for the extrinsic changes, we performed additional experiments and presented the data as three new Figs (**Extended data Fig. 8 and 9 and Fig. 7**) to demonstrate that the knockdown of Wnt4 was responsible for the changes of SCs after OSKM induction. Permanent Wnt4 knockout does not diminish the SC pool¹. It is possible the SC pool established a homeostasis similar to the one that is observed in muscles lacking Wnt4¹. Another possibility is that the SC homeostasis was not changed, as the SC niche was only temporarily modified. We discussed these points in the Discussion section.

3- Expression Timing: The authors show that the number of SC is increased in these mice in the absence of tissue injury, however it is not evaluated what is their fate, i.e. do they spontaneously contribute to myofibers? Is there an increase in the number of centrally nucleated and embryonic myosin heavy chain+ myofibers in these mice in the absence of injury? In addition, as they emerge from quiescence before the injury is performed, the improved tissue repair might be the result of this early activation, compounded with the potential ongoing contribution to myofibers. What happens if the expression of the Yamanaka factors is induced concurrently with tissue injury?

Response: We thank the reviewer for the comments. We analyzed the muscles after OSKM induction and we did not observe the myofibers with central nuclei (**Extended Data Fig. 5a**). In addition, immunostaining showed no eMHC⁺ myofibers in Cre⁺ muscles before the injury (**Extended Data Fig. 5b**), indicating that the activated SCs did not spontaneously contribute to myofibers of Cre⁺ muscles. However, there are two instances that may induce the activation of

SCs and spontaneous contribution to myofibers. The first is the change of intrinsic modulators of SCs that can induce their precocious differentiation. The other is the extrinsic signal (e.g., muscle degeneration), which can trigger SCs to enter myogenesis for muscle regeneration. Nevertheless, our myofiber-specific cyclic OSKM induction did not directly change the intrinsic modulators of SCs (**Fig. 2b**) or induce the degeneration of myofibers (**Extended Data Fig. 5a**); therefore, the SCs did not spontaneously activate and contribute to myofibers.

As the reviewer mentioned, the timing of OSKM induction and the early activated-SCs are important for accelerated-muscle regeneration. As suggested by the reviewer, we tested the effect of OSKM induction post-injury (**Extended Data Fig. 4**). We found that the induction of OSKM post-injury did not accelerate muscle regeneration. As the injury is a strong signal to trigger SCs activation and proliferation, the effect of OSKM on SCs activation and proliferation could be overwhelmed by the injury.

4- Molecular mechanism: A causal relationship between p21, Wnt4 and the observed phenotype is not formally demonstrated. The authors should perform ChIP-qPCR to validate that p21 directly binds Wnt4a promoter in myofibers, as well as luciferase reporter assays with deletion mutants for the p21 binding region on the Wnt4 promoter, to demonstrate causality of the binding to the increased Wnt4a transcription. In addition, that Wnt4a downregulation is mediating the observed phenotype should be established by rescue experiments, in which Wnt4a is upregulated or delivered in these mice. Finally, a discussion of the caveats should be included in the text, i.e. the observed phenotype could be either the result of a direct effect of myofibers on SC or an indirect effect through changes in other cell types in the tissue microenvironment. As several secreted factors (for example, Wnt4 could be upregulated also in other local cell types) or cell surface proteins could contribute to the phenotype.

Response: We again thank the reviewer for this important comment. As suggested by the reviewer, we performed ChIP-qPCR to validate that the TATA-proximal region of *Wnt4* promoter was enriched by p21 (**Fig. 6h**). We also designed a luciferase assay to verify that the TATA-proximal region is responsible for p21-mediated inhibition (**Fig. 6h**). Moreover, we did a rescue experiment by adding recombinant Wnt4 to block the changes of SCs on OSKM-overexpressing myofibers ex vivo to verify the knockdown of Wnt4 is responsible for SCs activation and expansion (**Extended data Fig. 8**). In addition, we utilized CasRx system to prove that cyclic Wnt4 knockdown could recapitulate OSKM-induced changes of SCs in vivo (**Fig. 7 and Extended data Fig. 9**).

We proved that expressing OSKM in myofibers induced p21, which inhibited *Wnt4*. The downregulation of *Wnt4* contributed to the activation of MyoD and Yap in SCs, resulting in their activation and expansion. We believe that the observed phenotype is primarily a direct effect of myofibers on SCs. However, we agree with the reviewer that the observed phenotype may result from an indirect effect through changes in the other component of the SC niche (e.g. ECM), as myofibers could secrete exosome to regulate the other cells². We added this point in the Discussion.

Specific points:

1- In Figure 3 the authors show that in the absence of injury the number of SC are increased ~2 fold in the tissue. However, upon tissue injury their number is increased only ~30%, which might suggest that cells are not being expanded but rather rapidly differentiating. Can the authors reconcile these two results?

Response: The SCs definitely expanded. The number of SCs was increased by 4-fold in Cre+ muscles and by 2-fold in Cre- muscle 3 days after CTX injection (**Fig. 2e and Fig. 3e**). During the early regeneration phase, SCs proliferate to increase their number and we could see the enlarged difference of the SC numbers (**Fig. 2e**). During the later regeneration stage, proliferated myoblasts will differentiate and fuse to myofibers, which results in a decrease of their number. During efficient muscle regeneration, a higher rate of fusion of SCs to the damaged myofibers take place, which could be a reason to flat the difference of SC number.

2- Quantification of western blot analysis in Figure 4F should be included to evaluate whether the difference in protein levels for p53 and p21 is statistically significant.

Response: We added the quantification of the western blots in Fig. 6f.

3- Figure S6c is unclear, it appears a repetition of experiments already shown in the manuscript, unclear if it is missing a panel here.

Response: We apologize for the confusion. We removed the panel.

4- Statistical analysis, N, P value and statistical method used should be included in all figure legends (it is missing for Figure 1).

Response: The data shown in Fig. 1c and d are from two replicates; therefore, we did not perform statistical analysis. However, the inductions of OSKM in the muscle were verified in **Fig. 2a**.

Reviewer #2 (Remarks to the Author):

In this manuscript, Wang, et al., reports that myofiber-specific, short-term (2 days), cyclic expression of Yamanaka reprogramming factors (OSKM) in young adult mice enables faster muscle regeneration upon injury due to increased number of satellite cells (SCs) via down-regulation of Wnt4, that maintain SCs quiescent. This is a follow up study of their previous publication (Ocampo et al., 2016), in which they demonstrated that ubiquitous short-term cyclic expression of OSKM ameliorates hallmarks of ageing, including the increased regeneration capacity of muscle in 12 months old mice. The phenotype, reported as if it was caused by epigenetic remodeling during cellular reprogramming, was quite surprising and very interesting. I welcome this new work, in which they demonstrated that at least the enhanced muscle regeneration capacity by OSKM in young mice could be explained by the increased SC number preceding the injury, while I think there are several key questions to be addressed in association with the previous work as below.

1. What % of muscle fibres become GFP+ (successful Cre excision) and what % can express OSKM? It is not clear if all muscle fibres can induce OSKM expression. Even if it is not 100%, is it sufficient to explain the phenotype? Immunofluorescence for Oct4 (and perhaps p21), images of GFP expression upon dox administration would be informative.

Response: We thank the reviewer for this important comment. We performed immunostaining for GFP and Sox2 in the sections of EDL muscles. As shown in **Extended data Fig. 1**, GFP and Sox2 signals were only found in Cre+ muscles; both GFP (in cytoplasm) and Sox2 (in myonuclei) signals were generally located in the same myofiber, with about 5% exceptions (GFP⁺Sox2⁻ or GFP⁻Sox2⁺). We quantified the Sox2⁺ myofibers, and there were 78% myofibers with Sox2⁺ myonuclei in EDL muscles (**Extended data Fig. 1**). Moreover, we also performed immunostaining of Oct4 and Pax7 in the sections of TA muscles (**Fig. 2b**). We found a profound induction of Oct4 in myonuclei of Cre+ myofibers, but not in Cre- myofibers or SCs (Pax7+ cells). These results show that GFP, Oct4 and Sox2 signals are seen in the majority of myofibers of Cre+ muscles.

2. Does the number of SCs increase even when OSKM are expressed systemically (i.e., in all cell types, including muscle fibre and SCs)? I expect SCs also up-regulate p21 expression in response to OSKM expression. Doesn't it inhibit SC proliferation? Or SCs do not up-regulate p21? Or the previous system did not have strong OSKM expression in SCs?

Response: We thank the reviewer for raising these important points. In the systemic-OSKM model, the level of OSKM in the skeletal muscles is much lower compared to the myofiber-specific or SC-specific OSKM model. Moreover, using the same induction condition (Dox in drinking water), we did not detect the upregulation of p21 in the skeletal muscles of systemic-OSKM mice. With a local injection of Dox in skeletal muscles, we observed the OSKM expression in the systemic-OSKM mice. However, the expression level of OSKM is 5-10 times lower in the systemic model compared to myofiber-specific (Fig. 1 and **Fig. 2a**).

Fig I. The levels of OSKM in skeletal muscles of systemic-OSKM mice after local Dox or PBS injection.

3. The observed phenotype in this work can be seen in old mice? i.e. the improved muscle regeneration capacity in old mice in the previous paper can be explained by the increased number of SCs (caused by suppression of Wnt4)?

Response: To address the reviewer’s question, we did immunostaining of Pax7 and MyoD in TA muscles of 15 month-old systemic 4F mice, which were injected with 3 cycles of Dox or PBS (as control) (Fig. II). We did not see a significant difference in the number of Pax7⁺ cells, but the percentage of activated (MyoD⁺) SCs was increased (Fig. II). There are two possible explanations for the unchanged SC proliferation. First, the induction of p21 should negatively regulate the G₁ to S progression of satellite cells³, so that p21 may inhibit the proliferation of SCs in the systemic 4F mice. Second, the OSKM level may not be high enough to drive the proliferation of SCs.

Fig II. The number and state of SCs in TA muscles of Systemic 4F mice.

Next, we investigated the status of SCs in 15 month-old myofiber-specific 4F mice. Acta-Cre/4F^{het} mice were administrated with 3 cycles of Dox. After the second and third rounds of Dox administration, BrdU was added to the drinking water for 2 days to label cycling SCs (Fig. IIIa). Cre⁺ muscles had 1.8-fold more SCs and 2.8-fold more MyoD⁺ SCs than Cre⁻ muscles (Fig. IIIb-d). Co-staining of BrdU and Pax7 showed that Cre⁺ muscles had a 3-fold higher percentage of BrdU⁺ SCs than Cre⁻ muscles (Fig. IIIe and f). We then studied muscle regeneration capacity 7 days post CTX injection after cyclic OSKM induction (Fig. IIIg). Cre⁺ muscles had less eMHC⁺ myofibers compared with Cre⁻ muscles (Fig. IIIh and i). In addition, Cre⁺ muscles had more SCs than Cre⁻ muscles (Fig. IIIj and k). Moreover, myofibers in Cre⁺ muscles were larger than those observed in Cre⁻ muscle, with more myofibers that were 30-40 μ m and fewer myofibers that were 10-20 μ m in diameter (Fig. IIIl). Together, these results indicate that myofiber-specific OSKM induction accelerates SC expansion and muscle regeneration in aging mice. We do not think the improved muscle regeneration capacity in our previous paper is due to the increased number of SCs, as there is no change in the number of SCs in the systemic 4F mice without injury (Fig. II).

Fig III. Myofiber-specific OSKM induction accelerates muscle regeneration in aging mice.

a, Schematic representation of the experimental design. **b-d**, Immunostaining of Pax7 and MyoD and quantification of Pax7⁺ cells per field and the percentage of MyoD⁺ SCs in TA muscle sections. Pax7⁺ cells are indicated by arrows. Scale bars= 50 μ m. **e** and **f**, Immunostaining of Pax7 and BrdU and the quantification of the percentage of BrdU⁺ SCs in TA muscle sections. Scale bars= 10 μ m. **g**, Schematic representation of the experimental design. **h** and **i**, Immunostaining of embryonic myosin heavy chain (eMHC) and Dystrophin in TA muscle sections, and the quantification of the percentage of immature myofibers that express eMHC. Scale bars= 50 μ m. **j** and **k**, Immunostaining of Pax7 and MyoD, and quantification of Pax7 cells in TA muscle sections. Scale bars= 50 μ m. Representative regions are shown at higher magnification. Scale bars= 25 μ m. Arrows indicate Pax7⁺ cells. **l**, H&E staining of TA muscle sections and myofiber size distributions in TA muscle sections. Scale bars= 100 μ m. Error bars represent mean+s.d. A two-sided unpaired Student's t-test was performed.

4.50% reduction of Wnt4 (Fig 4c) is sufficient to explain the phenotype? Can addition of Wnt4 block SC proliferation in vitro, as shown in Figure S1G in <https://doi.org/10.1016/j.stem.2019.08.007> (Wnt4 from the Niche Controls the Mechano-Properties and Quiescent State of Muscle Stem Cells, Cell Stem Cell, 2019)?

Response: As suggested by the reviewer, we performed a rescue assay by adding recombinant Wnt4 to block the changes of SCs on OSKM-overexpressing myofibers ex vivo. The rWnt4 significantly decreased the expression of MyoD and Pax7⁺ cells on the myofibers 48h after treatment (**Extended data Fig. 8c-g**). These results indicate that the repression of Wnt4 was responsible for OSKM induction-induced SCs activation and expansion. Further, we utilized CasRx system to prove that cyclic Wnt4 knockdown could recapitulate OSKM-induced changes of SCs in vivo (**Fig. 7 and Extended data Fig. 9**).

5. Is it right to conclude that the increased regeneration capacity of muscle by OSKM expression in old mice (Ocampo, et al.,) was due to increased SC number, not due to epigenetic rejuvenation of the cells? While one might call 'increasing SCs' rejuvenation of muscle, it is not specific in old muscle. I believe it is important to clearly state that the increased regeneration capacity is a secondary effect of increased SC numbers, not due to magical rejuvenation of old cells (unless epigenetic rejuvenation is also important for old SCs and/or muscle to show this phenotype).

Response: In the systemic OSKM-mice, we did not see an increase of SC number before muscle injury, but the percentage of activated SCs was higher (Fig. II). Activated SCs can enter into myogenesis program faster than the quiescent SCs to accelerate muscle regeneration. Although we still do not know the possibilities of SCs rejuvenation or the role of other cell types (macrophages, FAPs, fibroblasts, etc.) affected by OSKM to help muscle regeneration in the systemic 4F mice. Our current manuscript raises a point that OSKM could improve muscle regeneration in a way independent of rejuvenation and opened a new angle to understand the role of OSKM in tissue regeneration. To better understand whether SC-specific or myofiber-specific OSKM induction could rejuvenate SCs and improve muscle regeneration requires further studies in ageing animals.

Minor points,

6. Did the authors sort GFP+ cells for the RNA-seq (if the excision efficiency is not high)?

Response: For the RNA-seq analysis, we did not sort GFP+ cells, as a high percentage (78%) of myofibers had GFP signal and showed OSKM induction.

7. Do you see increased SCs in the context of CAG-p21 electroporation (Fig 4g)?

Response: SCs are activated and expanded for a period after electroporation⁴; therefore, in the context of CAG-p21 electroporation, we were not able to demonstrate the role of p21 on SC proliferation.

Reviewer #3 (Remarks to the Author):

Previously the author's group reported that global over-expression of OSKM in mice is capable of inducing an expansion of muscle stem cells in older mice and improving muscle regeneration following injury. In this manuscript, the authors utilized muscle fiber-specific overexpression of OSKM in mice to examine whether the short-term OSKM-overexpressing niche has an effect on satellite cell (SC) function. As a result, OSKM-overexpressing induces SC activation and proliferation, increases in the number of SCs, and induces faster muscle regeneration. RNA-seq data showed increased expression of p21 and p53, and decreased expression of Wnt4 which recently showed an essential role in SC quiescence and a negative-regulator for SC differentiation, suggesting that short term OSKM-overexpressing may promote muscle regeneration by modifying the stem cell niche.

This is a follow-up manuscript after a recently published paper in which the global over-expression of OSKM can induce SC expansion and muscle regeneration in old mice. Overall, the mechanism for fiber-OSKM-mediated SC expansion is of interest, the molecular mechanism of this process is still preliminary.

Major issues:

1. OSKM-overexpressing can induce around 50% Wnt4 mRNA reduction. It is not so clear whether this reduction can induce SC expansion and enhanced muscle regeneration. The authors should address this question by Wnt4-overexpression, and Wnt4 gene knockdown and knockout mice.

Response: We thank the reviewer for the valuable suggestion. To validate that the reduction of Wnt4 was responsible for the SC activation and expansion, we did a rescue assay by adding recombinant Wnt4 to block the changes of SCs on OSKM-overexpressing myofibers ex vivo (**Extended data Fig. 8**). The rWnt4 significantly decreased the expression of MyoD and Pax7⁺ cells on the myofibers 48h after treatment (**Extended data Fig. 8c-g**). These results indicate that the repression of Wnt4 was responsible for OSKM induction-induced SCs activation and expansion. Using Wnt4 knockout mice, Andrew Brack's group demonstrated that knockout of Wnt4 induces the activation and proliferation of SCs¹. In our study, we utilized CasRx system to knockdown Wnt4 and recapitulated OSKM-induced changes of SCs in vivo. The results from this new experiment are presented as new figures in **Fig. 7 and Extended data Fig. 9**.

2. Overall, muscle regeneration data is very preliminary. Please show the earlier phase as well as later time course after CTX injection. For example, the authors should examine day 2 or 3 for EdU/Pax7 staining and eMHC staining. In addition, day 14 and later stages for HE staining and fibrosis staining to see whether muscle regeneration is caught up at the later

stages in OSKM-mice. Finally, it would be interesting to see whether OSKM-mice have better muscle function.

Response: We thank the reviewer for raising this important point. As suggested by the reviewer, we performed BrdU/Pax7 co-staining and eMHC staining on day 3 (**Fig. 2c-e**). Similarly, we added MyoD/Pax7, eMHC, H&E and fibrosis staining's at 14 (**Fig. 4a-f**) and 30 days after CTX injection (**Fig. 4g-l**). The muscle regeneration was still better in Cre+ muscles than Cre- muscles till day 14. Moreover, we also studied the muscle function by measuring the grip strength of hinder limbs 0, 1 and 6 days post CTX injection. The Cre+ muscles have higher grip strength than Cre- muscles 6 days after CTX injury (**Fig. 2j**).

3. Since *Wnt4* is important for SC-quiescence. It would be nice to try multiple CTX-mediated muscle injury in OSKM mice to see whether SC-quiescence is reduced by OSKM overexpression and muscle regeneration capacity is diminished after multiple regeneration cycles.

Response: We again thank the review for this great suggestion. We investigated the state and number of SCs and muscle regeneration capacity after three times CTX injection with 7 days interval between each injection (**Fig. 5**). Our results show that the number of Pax7⁺ cells were comparable in Cre+ and Cre- muscles 7 and 30 days after triple CTX injection. Similarly, the percentage of MyoD⁺Pax7⁺ cells were comparable in Cre+ and Cre- muscles 30 days after triple CTX injection. In addition, the muscle regeneration capacity was normal in Cre+ muscle after triple CTX injection. These results prove that the self-renewal of SCs was maintained after OSKM induction.

Minor issues:

1. In Figure 3, it would be nice to show OSKM protein expression on muscle fibers.

Response: As suggested by the reviewer, we added the immunostaining of Sox2 in sections of EDL muscles (**Extended data Fig. 1**). We also added the immunostaining of Oct4 and Pax7 in sections of TA muscles (**Fig. 2b**). We see a profound induction of Oct4 in myonuclei of Cre+ muscle fibers, but not in Cre- myofibers or SCs (Pax7⁺ cells).

2. In Figure 3, please show the ration of Pax7⁺, MyoD⁺ and Pax7⁺MyoD⁺ in single muscle fiber experiments.

Response: We added the ratios (**Fig. 3f**).

3. In Figure 4i, please perform MyoD/YAP double staining and quantitative analysis on muscle fibers.

Response: As per the suggestion, we added the MyoD/active Yap double staining and quantifications in **Extended data Fig. 8**.

References

1. Eliazar, S. *et al.* Wnt4 from the Niche Controls the Mechano-Properties and Quiescent State of Muscle Stem Cells. *Cell stem cell* **25**, 654-665 e654 (2019).
2. Fry, C.S., Kirby, T.J., Kosmac, K., McCarthy, J.J. & Peterson, C.A. Myogenic Progenitor Cells Control Extracellular Matrix Production by Fibroblasts during Skeletal Muscle Hypertrophy. *Cell stem cell* **20**, 56-69 (2017).
3. McCroskery, S., Thomas, M., Maxwell, L., Sharma, M. & Kambadur, R. Myostatin negatively regulates satellite cell activation and self-renewal. *Journal of Cell Biology* **162**, 1135-1147 (2003).
4. Aihara, H. & Miyazaki, J. Gene transfer into muscle by electroporation in vivo. *Nature biotechnology* **16**, 867-870 (1998).

REVIEWER COMMENTS

Reviewer #1 (Remarks to the Author):

The authors have addressed previous concerns by performing several additional experiments, and as a result the resubmitted manuscript is substantially strengthened. I only have the following remaining comments:

1- Expansion: As the work shows that OSKM induces muscle stem cell activation and expansion of myogenic progenitors (Pax7+MyoD+), not muscle stem cell expansion (Pax7+MyoD-), I would suggest to change the sentence in the abstract "Here, we demonstrate that the expression of OSKM, specifically in myofibers, induced the expansion of muscle stem cells or satellite cells (SCs), which accelerated muscle regeneration in young mice.", which is misleading. Further, the authors define Pax7+MyoD- cells as quiescent muscle stem cells, while they could also be self-renewing muscle stem cells. These two aspects and terminologies should be modified throughout the text, for clarity.

2- Figures 3M: While the work shows that OSKM induces muscle stem cell activation and expansion of myogenic progenitors (Pax7+MyoD+), in Figure 3M, after 3d in culture, they only show more Pax7+ cells per cluster, and they conclude "Collectively, these results prove that myofiber-specific expression of OSKM induced the activation and expansion of SCs.", while they should also show MyoD quantification (as they show the staining), which, based on results shown in the rest of figure 3, should account for the difference. This would be important for the audience, to clearly interpret these findings, i.e. the treatment promotes muscle stem cell activation and expand Pax7+MyoD+ committed progenitors, not muscle stem cells. The same for Figures 2I and 7H.

3- In Figure 2D the staining for eMyHC is not convincing, as all fibers shown in both Cre- and Cre+ samples appear to have a diffuse red fluorescence. Please include more representative images of the quantification shown.

Reviewer #2 (Remarks to the Author):

The authors answered well to most of my concerns, but I have the following comments.

One of my main questions was if the mechanism of the increased regeneration capacity of skeletal muscle-specific induction of OSKM in the current manuscript can explain the rejuvenation AND increased regeneration capacity of old mice by systemic OSKM expression in their previous paper. I understand the authors do not think so because 1. OSKM expression in the systemic inductions system is very low (Fig. I), and 2. the number of Pax7+ BrdU+ cells per field is not increased by systemic OSKM expression (Fig. II).

Regarding Fig.I, I wonder why the authors did not show me OSKM expression under the same induction condition (Dox in drinking water), but used local Dox injection. Is that the OSKM induction was too low to detect with the same condition?

Regarding Fig.II, I see the difference of the number of Pax7+ BrdU+ cells per field is not statistically significant ($p=0.22$). However, the difference is not that different from other figures (e.g. Figs 2i, 3e, Fig IIIc), while the error bars are bigger and the sample numbers are smaller in Fig.II ($n=3$). So, I am not sure the conclusion that 'We did not see a significant difference in the number of Pax7+ cells' is correct if the authors have the similar sample numbers to other figures. In addition, I wonder why the

Y axis of 'Pax7+ cells per field' graphs are so different in each figures (5 to 50). Is that because the size of the 'field' is different in each figure or the number of Pax7+ cells are very different depending on where you image? If it is the latter, it's a big problem throughout the manuscript because the ~2-fold difference in 'Pax7+ cells per field' might not mean anything.

As for one of the explanation of the unchanged SC proliferation in Fig II 'the induction of p21 should negatively regulate the G1 to S progression of satellite cells'. The authors described that p21 upregulation is not detectable in the skeletal muscles of systemic-OSKM mice (related to Fig.I). Do the authors see p21 up-regulation in the satellite cells of systemic-OSKM mice? Even though the OSKM expression is very low?

I understand that the authors' point of this manuscript is that 'OSKM could improve muscle regeneration in a way independent of rejuvenation', thus the above my concerns are not major points (except the Y axis of Pax7+ cell number graphs). But the current abstract sounds like that this work addressed mechanisms of OSKM-mediated increased regeneration shown in their previous paper. My understanding is that the increased regeneration capacity demonstrated in this manuscript is totally different from the 'rejuvenation AND increased regeneration capacity' in their previous paper, due to very different OSKM expression levels, while myofiber-specific high level OSKM induction might also rejuvenate SCs, and which requires further studies in ageing animals. I do not demand further large experiments, but I believe this manuscript should clearly state that the mechanism described here is different from what is shown in the previous paper, demonstrating the difference of OSKM expression levels and lack of p21 induction in skeletal muscle in the systemic-OSKM mice in in the figure.

Prof Keisuke Kaji
Biology of Reprogramming
Centre for Regenerative Medicine
Institute for Regeneration and Repair
University of Edinburgh

Reviewer #3 (Remarks to the Author):

Overall, the authors well responded to the reviewers' comments. I have some questions before publication. It is not still clear how OSKM can upregulate p21 expression. Please comment on this molecular mechanism. Upregulation of p21 usually induces G1 arrest. However, it seems not to be happening in the OSKM-overexpressing muscle. Please explain this phenomenon.

Response to Reviewers' Comments

We thank the reviewers for their recognition of our efforts to strengthen the manuscript and appreciate their additional suggestions. We have addressed the comments and suggestions from the reviewers in our revised manuscript. Please see below our point-to-point responses.

Reviewer #1 (Remarks to the Author):

The authors have addressed previous concerns by performing several additional experiments, and as a result the resubmitted manuscript is substantially strengthened. I only have the following remaining comments:

1- Expansion: As the work shows that OSKM induces muscle stem cell activation and expansion of myogenic progenitors (Pax7+MyoD+), not muscle stem cell expansion (Pax7+MyoD-), I would suggest to change the sentence in the abstract "Here, we demonstrate that the expression of OSKM, specifically in myofibers, induced the expansion of muscle stem cells or satellite cells (SCs), which accelerated muscle regeneration in young mice.", which is misleading. Further, the authors define Pax7+MyoD- cells as quiescent muscle stem cells, while they could also be self-renewing muscle stem cells. These two aspects and terminologies should be modified throughout the text, for clarity.

Response: Thank you to the reviewer for this suggestion. We agree that our previous sentence was misleading. We have changed this sentence to "Here, we demonstrate that the expression of OSKM, specifically in myofibers, induced the activation of muscle stem cells or satellite cells (SCs), which accelerated muscle regeneration in young mice." We defined Pax7⁺MyoD⁻ cell as SCs and Pax7⁺MyoD⁺ cells as myoblasts¹. We agree with the reviewer that Pax7⁺MyoD⁻ SCs are quiescent muscle stem cells that are capable of self-renewal. We modified the definition throughout the text to avoid confusion between the terms.

2- Figures 3M: While the work shows that OSKM induces muscle stem cell activation and expansion of myogenic progenitors (Pax7+MyoD+), in Figure 3M, after 3d in culture, they only show more Pax7+ cells per cluster, and they conclude "Collectively, these results prove that myofiber-specific expression of OSKM induced the activation and expansion of SCs.", while they should also show MyoD quantification (as they show the staining), which, based on results shown in the rest of figure 3, should account for the difference. This would be important for the audience, to clearly interpret these findings, i.e. the treatment promotes muscle stem cell activation and expand Pax7+MyoD+ committed progenitors, not muscle stem cells. The same for Figures 2I and 7H.

Response: We appreciate the reviewer's suggestion to add the MyoD quantification, which will help us to distinguish Pax7⁺MyoD⁻ SCs and Pax7⁺MyoD⁺ myoblasts. As all Pax7 cells are

MyoD⁺ in Fig. 3m, we modified the y-axis label by changing Pax7⁺ cells to Pax7⁺MyoD⁺ myoblasts. For Fig. 2i and 7h, we qualified Pax7⁺MyoD⁻ SCs per field and Pax7⁺MyoD⁺ myoblasts per field. We also revised the text accordingly to clarify the interpretation of our findings.

3- In Figure 2D the staining for eMyHC is not convincing, as all fibers shown in both Cre- and Cre+ samples appear to have a diffuse red fluorescence. Please include more representative images of the quantification shown.

Response: As you can see in the additional representative images (Fig. A), staining background (diffuse red fluorescence) was found in degenerated myofibers (lack of Dystrophin staining). Degenerated myofibers were permeable to mouse IgG. As we used an anti-mouse IgG secondary antibody to detect eMHC (mouse IgG1), the uptake of mouse IgG caused the staining background (diffuse red fluorescence). We only counted the central-nucleated myofibers with strong (relative to background) eMHC signals. To avoid potential confusion, we added arrows to indicate the eMHC⁺ myofibers in Fig. 2D.

Figure A. Immunostaining of eMHC, Dystrophin and DAPI in muscles sections 3 days post CTX injection

Reviewer #2 (Remarks to the Author):

The authors answered well to most of my concerns, but I have the following comments.

One of my main questions was if the mechanism of the increased regeneration capacity of skeletal muscle-specific induction of OSKM in the current manuscript can explain the rejuvenation AND increased regeneration capacity of old mice by systemic OSKM expression in their previous paper. I understand the authors do not think so because 1. OSKM expression in the systemic inductions system is very low (Fig. I), and 2. the number of Pax7+ BrdU+ cells per field is not increased by systemic OSKM expression (Fig. II).

Response: Thank you to the reviewer for this comment. In our previous paper, we found that skeletal muscle regeneration was improved with OSKM induction in systemic 4F mice. However, we did not know whether the skeletal muscle was rejuvenated. Our current study could not explain the improved regeneration capacity in systemic 4F mice. This is due to the fact that SCs are regulated by not only myofibers, but also macrophages, FAPs, fibroblasts, etc². Our current study proves that OSKM induction in myofiber can improve muscle regeneration by inducing the early activation of SCs. However, we could not draw a conclusion about systemic 4F mice without knowing the contributions of the other types of cells regulating SCs after OSKM induction in systemic 4F mice. This is a very interesting point that will be addressed in detailed in a future study and we thank this reviewer for pointing it out.

Regarding Fig.I, I wonder why the authors did not show me OSKM expression under the same induction condition (Dox in drinking water), but used local Dox injection. Is that the OSKM induction was too low to detect with the same condition?

Response: Yes, we did not observe OSKM induction in skeletal muscle of systemic 4F mice with Dox in drinking water (Fig. B).

Figure B. The mRNA level of OSKM in skeletal muscles of systemic-4F mice after feeding with water containing PBS or Dox.

Regarding Fig.II, I see the difference of the number of Pax7+ BrdU+ cells per field is not statistically significant ($p=0.22$). However, the difference is not that different from other figures (e.g. Figs 2i, 3e, Fig IIIc), while the error bars are bigger and the sample numbers are smaller in Fig.II ($n=3$). So, I am not sure the conclusion that 'We did not see a significant difference in the number of Pax7+ cells' is correct if the authors have the similar sample numbers to other figures.

Response: We agree with the reviewer that we may see a significant difference when we increase the sample numbers. However, we still do not know whether the change is solely a result from myofibers, since the other types of cells have the potential to regulate the proliferation of SCs in systemic 4F mice.

In addition, I wonder why the Y axis of 'Pax7+ cells per field' graphs are so different in each figures (5 to 50). Is that because the size of the 'field' is different in each figure or the number of Pax7+ cells are very different depending on where you image? If it is the latter, it's a big problem throughout the manuscript because the ~2-fold difference in 'Pax7+ cells per field' might not mean anything.

Response: To clarify, the size of the field is consistent in counting the number of Pax7⁺ cells. In addition, three to five different areas were analyzed for each section and three to four sections were analyzed for each sample to minimize the technical variance of the counting.

The big difference of Pax7+ cells per field mentioned by the reviewer is dependent on whether the samples were injured by CTX. The Pax7+ cells will proliferate to increase their number during muscle regeneration after CTX injury, so the number '50' was counted in samples 6 days after CTX injection, while the number '5' was counted in samples without CTX injury. We only compared the effect of OSKM induction on the same condition.

As for one of the explanation of the unchanged SC proliferation in Fig II 'the induction of p21 should negatively regulate the G1 to S progression of satellite cells'. The authors described that p21 upregulation is not detectable in the skeletal muscles of systemic-OSKM mice (related to Fig.I). Do the authors see p21 up-regulation in the satellite cells of systemic-OSKM mice? Even though the OSKM expression is very low?

Response: We did see the up-regulation of p21 in satellite cells of systemic-OSKM mice after local injection of Dox (Fig. C), but p21 was not upregulated in satellite cells of systemic-OSKM mice that were fed with water containing Dox (Fig. C).

Figure C. The mRNA level of p21 in satellite cells isolated from skeletal muscles of systemic-4F mice after feeding with water containing PBS or Dox or local injection of PBS or DOX.

I understand that the authors' point of this manuscript is that 'OSKM could improve muscle regeneration in a way independent of rejuvenation', thus the above my concerns are not major points (except the Y axis of Pax7+ cell number graphs). But the current abstract sounds like that this work addressed mechanisms of OSKM-mediated increased regeneration shown in their previous paper. My understanding is that the increased regeneration capacity demonstrated in this manuscript is totally different from the 'rejuvenation AND increased regeneration capacity' in their previous paper, due to very different OSKM expression levels, while myofiber-specific high level OSKM induction might also rejuvenate SCs, and which requires further studies in ageing animals. I do not demand further large experiments, but I believe this manuscript should clearly state that the mechanism described here is different from what is shown in the previous paper, demonstrating the difference of OSKM expression levels and lack of p21 induction in skeletal muscle in the systemic-OSKM mice in in the figure.

Response: We thank the reviewer for the comment and apologize for the confusion. We revised our abstract to clarify the message of the current manuscript. We agree with the reviewer that the current manuscript does not address the mechanism of rejuvenation observed in our previous study and appreciate that the reviewer understands that it requires the use of ageing animals. We stated this in the manuscript.

Reviewer #3 (Remarks to the Author):

Overall, the authors well responded to the reviewers' comments. I have some questions before publication. It is not still clear how OSKM can upregulate p21 expression. Please comment on this molecular mechanism. Upregulation of p21 usually induces G1 arrest. However, it seems not to be happening in the OSKM-overexpressing muscle. Please explain this phenomenon.

Response: Thank you for these comments. As for the molecular mechanism of p21 upregulation, we found the increase in p53 protein levels after OSKM induction, as previously reported³, and therefore p53 may be responsible for p21 induction although further analysis will be required. So far we did not find any p53-independent pathway for p21 induction and therefore we modified Fig. 6j.

In our myofiber-specific 4F model, 4F was only induced in myofibers but not in satellite cells (Fig. 2b and Extended Data Fig. 1). We verified that OSKM and p21 was not upregulated in satellite cells isolated from the myofiber-specific 4F model (Fig. D), thus the concern of G1-arrest was not applicable in the myofiber-specific 4F model.

Fig D. The mRNA levels of OSKM and p21 in satellite cells isolated from skeletal muscles of myofiber-specific 4F mice.

References

1. Kuang S, Gillespie MA, Rudnicki MA. Niche regulation of muscle satellite cell self-renewal and differentiation. *Cell stem cell* **2**, 22-31 (2008).
2. Yin H, Price F, Rudnicki MA. Satellite cells and the muscle stem cell niche. *Physiological reviews* **93**, 23-67 (2013).
3. Hong H, *et al.* Suppression of induced pluripotent stem cell generation by the p53-p21 pathway. *Nature* **460**, 1132-1135 (2009).

REVIEWER COMMENTS

Reviewer #2 (Remarks to the Author):

The abstract now sounds less likely to be a mechanistic analysis of their previous report. I also understand the Y axis scale of SC numbers are different in different Figures. Sorry for my lack of understanding.

While the author responded 'we could not draw a conclusion about systemic 4F mice without knowing the contributions of the other types of cells regulating SCs after OSKM induction in systemic 4F mice.', I understand that the authors stated that the regeneration enhancement mechanism in this paper is different from the one in the previous paper, because 'the expression level of OSKM is 5-10 times lower in the systemic model compared to myofiber-specific (Fig. I and Fig. 2a), and there is no change in the number of SCs in the systemic 4F mice without injury (Fig. II).'

I thought these two data and this conclusion were important to be included in this manuscript, as the readers can clearly distinguish these two important papers, and understand that there would be different rejuvenation mechanism(s) in the systemic 4F mice.

However, I also realized that in their previous paper the authors described that OSKM induction upon intramuscular injection of doxycycline into the TA muscle in the same 4F mice was 'strong' (Figure S7G). It is indeed 'strong' in the Figure S7G and quite different from Fig I (at least for Oct4 and Sox2). I am really sorry to realize it now, but I wonder why the OSKM induction after local Dox injection to the same systemic-OSKM mice is so different in the two papers (age difference?).

The previous paper also said 'Lastly, analysis of satellite cells revealed an increase in the number of Pax7-positive satellite cells in the muscle of 12-month-old 4F mice treated with doxycycline compared to untreated controls (following muscle injury by CTX injection) (Figure 7H).' This looks similar to what the authors observed in the current manuscript with myofiber-specific expression of OSKM. While Fig II with local dox injection into systemic-OSKM mice did not show a statistically significant increase of SC numbers (which the authors agreed that it might become statistically significant if they increase the sample number) in the absence of injury, it had low induction of OSKM unlike their previous paper. I do agree with the authors that 'we still do not know whether the change is solely a result from myofibers, since the other types of cells have the potential to regulate the proliferation of SCs in systemic 4F mice.', even if we see an increased number of SCs in the systemic 4F mice without injury (Fig. II). However, Fig III showed that regeneration enhancement with myofiber-specific expression of OSKM CAN also happen in aged mice. Thus, it CAN also be explained by solely myofibers.

I now wonder if the mechanism described in this paper explains the enhanced muscle regeneration in Figure 7D-H of their previous paper, which was in fact not by systemic OSKM induction. While I also understand that different cell types in the Dox injection sites can express OSKM, I guess the major cell type is muscle cells. Isn't it possible that the previous paper had two different phenotypes (systemic rejuvenation and local tissue regeneration that can also happen in young mice independently from epigenetic rejuvenation), and this paper explains the mechanism of the latter? If it is the case, nothing wrong with it but it important to clarify that possibility.

I really do not like to keep asking more experiments, but I feel this will change the message of 2 important papers.

Response to reviewers' comments

We appreciate the reviewer's suggestions that helped present clearly the message of our manuscript. We tried our best to address the comments and suggestions. Please see below our point-to-point responses.

The abstract now sounds less likely to be a mechanistic analysis of their previous report. I also understand the Y axis scale of SC numbers are different in different Figures. Sorry for my lack of understanding.

While the author responded 'we could not draw a conclusion about systemic 4F mice without knowing the contributions of the other types of cells regulating SCs after OSKM induction in systemic 4F mice.', I understand that the authors stated that the regeneration enhancement mechanism in this paper is different from the one in the previous paper, because 'the expression level of OSKM is 5-10 times lower in the systemic model compared to myofiber-specific (Fig. I and Fig. 2a), and there is no change in the number of SCs in the systemic 4F mice without injury (Fig. II).'

I thought these two data and this conclusion were important to be included in this manuscript, as the readers can clearly distinguish these two important papers, and understand that there would be different rejuvenation mechanism(s) in the systemic 4F mice.

Response: Thank you for your suggestions. We now incorporated the two figures into the manuscript as Extended Fig. 9.

However, I also realized that in their previous paper the authors described that OSKM induction upon intramuscular injection of doxycycline into the TA muscle in the same 4F mice was 'strong' (Figure S7G). It is indeed 'strong' in the Figure S7G and quite different from Fig I (at least for Oct4 and Sox2). I am really sorry to realize it now, but I wonder why the OSKM induction after local Dox injection to the same systemic-OSKM mice is so different in the two papers (age difference?).

Response: We thank the reviewer for raising this point. We used the same primers and included more samples to replicate the gene expression analysis. Mice age is similar to the previous study (around 12 months old). When we presented the data in Transcript levels/Gapdh, we found the induction levels of OSKM are similar to those observed in our previous study (Fig. A). However, the background levels are higher for Oct4 and Sox2 in our current data compared to the previous study (Fig. A). The only procedure difference is that we are currently using a sonicator to homogenize the muscle tissue, whereas mechanical homogenization (without sonication) was done in the previous study. We found that sonication could greatly increase the quantity and quality of RNA from skeletal muscle compared to mechanical homogenization and could largely

increase the background levels of *Oct4* and *Sox2*. We included this data in Extended Fig. 9, which also contain the OSKM level (expressed as Transcript levels/Gapdh) of myofiber-specific mice to show the OSKM expression levels in both models.

Fig. A The expression level of OSKM in systemic 4F mice after Dox or PBS treatments.

The previous paper also said ‘Lastly, analysis of satellite cells revealed an increase in the number of Pax7-positive satellite cells in the muscle of 12-month-old 4F mice treated with doxycycline compared to untreated controls (following muscle injury by CTX injection) (Figure 7H).’ This looks similar to what the authors observed in the current manuscript with myofiber-specific expression of OSKM. While Fig II with local dox injection into systemic-OSKM mice did not show a statistically significant increase of SC numbers (which the authors agreed that it might become statistically significant if they increase the sample number) in the absence of injury, it had low induction of OSKM unlike their previous paper. I do agree with the authors that ‘we still do not know whether the change is solely a result from myofibers, since the other types of cells have the potential to regulate the proliferation of SCs in systemic 4F mice.’, even if we see an increased number of SCs in the systemic 4F mice without injury (Fig. II). However, Fig III showed that regeneration enhancement with myofiber-specific expression of OSKM CAN also happen in aged mice. Thus, it CAN also be explained by solely myofibers.

I now wonder if the mechanism described in this paper explains the enhanced muscle regeneration in Figure 7D-H of their previous paper, which was in fact not by systemic OSKM induction. While I also understand that different cell types in the Dox injection sites can express OSKM, I guess the major cell type is muscle cells. Isn’t it possible that the previous paper had two different phenotypes (systemic rejuvenation and local tissue regeneration that can also happen in young mice independently from epigenetic rejuvenation), and this paper explains the mechanism of the latter? If it is the case, nothing

wrong with it but it important to clarify that possibility.

I really do not like to keep asking more experiments, but I feel this will change the message of 2 important papers.

Response: Thank you for your insight and suggestions. We increased the sample number and reanalyzed previous samples from Fig. II. We combined Fig. I and Fig. II as Extended Fig. 9. In addition, we added Fig. III as Extended Fig. 8. With these new data, we have introduced a new section in the manuscript. We compared and explained the SCs changes in systemic 4F mice and myofiber-specific 4F mice and also mentioned the possible mechanisms contributing to the improved muscle regeneration in systemic 4F mice. Below is the new paragraph that we added to the manuscript.

Myofiber-specific OSKM induction consistently promotes activation and proliferation of SCs in aging mice

It is important to determine whether the effects of myofiber-specific OSKM induction on SCs are preserved in aging mice. We thus investigated the status of SCs in 15 month-old myofiber-specific 4F mice. Acta-Cre/4F^{het} mice were administrated with 3 cycles of Dox. After the second and third rounds of Dox administration, BrdU was added to the drinking water for 2 days to label cycling SCs (Extended Data Fig. 8a). Cre⁺ muscles had 1.8-fold more Pax7⁺ cells and 2.8-fold more activated (MyoD⁺) SCs than Cre⁻ muscles (Extended Data Fig. 8b-d). Co-staining of BrdU and Pax7 showed that Cre⁺ muscles had a 3-fold higher percentage of BrdU⁺ SCs than Cre⁻ muscles (Extended Data Fig. 8e and f). We then studied muscle regeneration capacity 7 days post CTX injection after cyclic OSKM induction (Extended Data Fig. 8g). Cre⁺ muscles had fewer eMHC⁺ myofibers compared with Cre⁻ muscles (Extended Data Fig. 8h and i). In addition, Cre⁺ muscles had more SCs than Cre⁻ muscles (Extended Data Fig. 8j and k). Moreover, myofibers in Cre⁺ muscles were larger than those observed in Cre⁻ muscle, with more myofibers that were 30-40 μm and fewer myofibers that were 10-20 μm in diameter (Extended Data Fig. 8l). Together, these results indicate that myofiber-specific OSKM induction promotes SC activation and proliferation and accelerates muscle regeneration independent of aging.

We previously showed that systemic OSKM induction could promote muscle regeneration in 12 month-old systemic 4F mice⁸. We decided to compare the changes of SCs before CTX injection in systemic 4F mice and Acta-Cre/4F^{het} mice. We did immunostaining of Pax7 and MyoD in TA muscles of 12 to 15 month-old systemic 4F mice, which were injected with 3 cycles of Dox or PBS (as control) according to our previous study⁸ (Extended Data Fig. 9a). We found that the percentage of activated (MyoD⁺) SCs was increased by 1.7-fold (Extended Data Fig. 9b and c). Activated SCs may enter into the myogenesis program faster than the quiescent SCs to accelerate muscle regeneration. The SC activation could be a result of the OSKM induction in myofibers. At the same time, we could not exclude the possibility that other cell types (macrophages, FAPs, fibroblasts, etc.) also contribute to the activation of SCs. We did not observe a significant

difference in the number of Pax7⁺ cells as seen in Acta-Cre/4F^{het} mice (Extended Data Fig. 9b and d). To analyze the differences, we first compared the induction level of OSKM. We found that the induction level of OSKM in the skeletal muscles of systemic 4F mice was lower compared to myofiber-specific 4F mice. With doxycycline treatment in drinking water, we did not consistently observe the induction of OSKM in the skeletal muscles of systemic-4F mice (data not shown). Therefore, we performed a local injection of Dox (1mg/ml) to induce the expression of OSKM in skeletal muscles of systemic 4F mice. The expression level of OSKM was 5-10 times lower than Acta-Cre/4F^{het} mice, which were administrated with doxycycline in water (Extended Data Fig. 9e and f). In addition, p21 was induced in the SCs of systemic 4F muscles (Extended Data Fig. 9g). The induction of p21 could negatively regulate the G₁ to S progression of satellite cells¹⁹. Our results provide an additional explanation for the improved muscle regeneration in systemic 4F mice besides SC rejuvenation.